# Progressive remote memory decline coincides with parvalbumin interneuron hyperexcitability and enhanced inhibition of cortical engram cells in a mouse model of Alzheimer's disease

Julia J van Adrichem, Rolinka J van der Loo, Romina Ambrosini Defendi, August B Smit, Michel C van den Oever*[†], Ronald E van Kesteren*[†]

Department of Molecular and Cellular Neurobiology, Center for Neurogenomics and Cognitive Research, Amsterdam Neuroscience, Vrije Universiteit Amsterdam, Amsterdam, Netherlands

*For correspondence:
michel.vanden.oever@vu.nl
(MCvdO);
ronald.van.kesteren@vu.nl
(REvK)

[†]These authors contributed
equally to this work

Competing interest: The authors
declare that no competing
interests exist.

Reviewing Editor: Matthew J
Rowan, Emory University, United
States

## eLife Assessment

This study provides **valuable** insights into the mechanisms of remote memory impairment in an Alzheimer's disease mouse model. The evidence is **compelling**, with careful use of viral-TRAP labeling and patch-clamp electrophysiology to demonstrate altered inhibitory microcircuit function, though the mechanistic link to memory deficits remains correlative. Overall, the work advances understanding of early circuit-level changes in AD, while highlighting open questions regarding causality and broader network contributions.

**Abstract** Patients with Alzheimer's disease (AD) initially show temporally graded retrograde amnesia, which gradually progresses into more severe retrograde amnesia. Although mouse models of AD have provided insight into neurobiological mechanisms contributing to impaired formation and retrieval of new memories, the process underlying the progressive loss of remote memories in AD has remained elusive. Here, we demonstrate age-dependent remote memory decline in APP/PS1 mice, which coincides with progressive hyperexcitability of parvalbumin (PV) interneurons in the medial prefrontal cortex (mPFC). Analysis of Fos expression showed that the remote memory deficit is not mirrored by changes in reactivation of memory-encoding neurons, so-called engram cells, nor PV interneuron (re)activation, in the mPFC. However, inhibitory input is enhanced onto engram cells compared to non-engram cells specifically in APP/PS1 mice. Our data indicate that age-dependent remote memory impairment in APP/PS1 mice is due to increased innervation of cortical engram cells by hyperexcitable PV interneurons, suggesting that dysfunctional inhibitory microcircuits in the neocortex mediate progressive retrograde amnesia in AD.

## Introduction

Alzheimer's disease (AD) is the most prevalent form of dementia and is characterized by amnesic cognitive impairment (*Knopman et al., 2021*). Specifically, patients with AD initially experience anterograde and temporally graded retrograde amnesia, i.e., an inability to form new memories and retrieve recently acquired memories (hours-to-days old). As the disease progresses, memory

impairments become more profound, resulting in severe retrograde amnesia, involving the loss of more remote memories (months-to-years old) (*El Haj et al., 2015*). The pathological hallmarks of the disease, neuritic plaques containing amyloid beta and neurofibrillary tangles composed of hyperphosphorylated tau, have been extensively studied to develop therapeutic interventions. Although some amyloid-targeting therapies are efficient in reducing the pathology and have recently been approved for the treatment of AD, cognitive effects of such treatments remain minimal or absent (*Avgerinos et al., 2024*; *Li et al., 2023*). Since memory loss is one of the earliest symptoms in AD and the most dehumanizing, it is important to understand its underlying mechanisms to identify new therapeutic targets that can also improve cognition or delay cognitive decline.

Memories are stored in the brain by sparse populations, or ensembles, of so-called engram cells (*Josselyn and Tonegawa, 2020*). These learning-activated neurons undergo structural, physiological, and molecular changes to encode, store, and retrieve specific memories (*Josselyn et al., 2015*). While initial formation and retrieval of contextual memories is hippocampus-dependent (*Liu et al., 2012*), memory persistence depends on memory retrieval by neocortical regions (*Frankland and Bontempi, 2005*). Specifically, engram cells in the medial prefrontal cortex (mPFC) are necessary for remote (>2-week-old), but not recent (<1-week-old), contextual memory expression in mice (*Kitamura et al., 2017*; *Matos et al., 2019*). The recruitment of neurons into an engram ensemble depends on their intrinsic excitability at the moment of learning, i.e., neurons with relatively higher excitability have a higher chance to become part of an engram (*Yiu et al., 2014*; *Zhou et al., 2009*).

Both AD patients and mouse models of AD exhibit changes in neuronal excitability (*Celone et al., 2006*; *Dickerson et al., 2005*; *Palop et al., 2007*). Prior to cognitive symptoms, individuals genetically at risk for AD already show increased hippocampal activation compared to noncarriers (*Filippini et al., 2009*). Similarly, APP23xPS45 mice show early changes in hippocampal neuron excitability before the presence of amyloid plaques (*Busche et al., 2012*). These alterations in excitability could potentially affect memory allocation to neurons upon memory encoding and/or engram reactivation during memory retrieval and might thereby contribute to memory loss in AD. Indeed, APP/PS1 mice show deficits in recent memory retrieval (24 hr after training) prior to amyloid plaque onset (*Roy et al., 2016*). Interestingly, the same study reported that memory recall could be induced upon optogenetic activation of the hippocampal engram, pointing to a memory retrieval deficit in the absence of artificial engram stimulation. It remains unknown whether remote memory formation and retrieval are similarly affected, and if so, whether altered memory engram function in the mPFC is involved.

Inhibitory interneurons play an important role in memory processing and engram function. Parvalbumin (PV) and somatostatin (SST)-expressing interneurons determine the size of an engram ensemble via inhibition of surrounding (non-engram) neurons (*Morrison et al., 2016*; *Stefanelli et al., 2016*). Moreover, SST neurons in the mPFC have been implicated in the encoding and expression of fear memory (*Cummings et al., 2022*; *Cummings and Clem, 2020*). Of relevance to this is that APP/PS1 mice present an early hyperexcitability of PV interneurons in the hippocampal CA1 at 4 months of age, which precedes pyramidal cell hyperexcitability, as well as the presence of amyloid plaques. Interestingly, chronic chemogenetic inhibition of hyperexcitable PV neurons subsequently rescues spatial memory function and precludes pyramidal cell hyperexcitability (*Hijazi et al., 2020a*). Additionally, hippocampal SST cell dysfunction, characterized by axon loss and impairment of structural plasticity of dendritic spines, has been observed in APP/PS1 mice starting around 5 months of age (*Schmid et al., 2016*). These findings suggest that early interneuron dysfunction may contribute to network instability and memory deficits in AD.

Here, we investigated whether remote memory function and excitability of PV and SST interneurons are altered in the mPFC of APP/PS1 mice. Furthermore, we determined whether changes in neuronal excitability affect the size, cellular composition, and reactivation of the mPFC engram ensemble that supports memory persistence. We show that age-dependent remote memory decline in APP/PS1 mice coincides with progressive hyperexcitability of PV, but not SST, cells in the mPFC. This remote memory deficit is not mirrored by changes in the PV cell composition or reactivation of the engram ensemble in the mPFC. However, our data point to increased inhibitory PV cell input onto engram cells compared to non-engram cells specifically in APP/PS1 mice, suggesting that remote memory engrams in the mPFC are impaired in APP/PS1 mice as a result of altered inhibitory synaptic transmission in engram cells, rather than changes in engram composition or size.

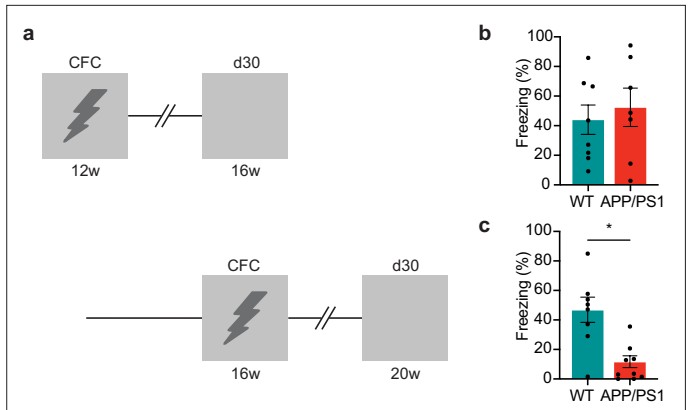

**Figure 1.** Progressive remote memory decline in APP/PS1 mice. (**a**) Experimental design. WT and APP/PS1 mice underwent contextual fear conditioning (CFC) at 12 weeks and 16 weeks of age, and memory retrieval 30 days later at 16 weeks and 20 weeks of age, respectively. (**b**) At 16 weeks of age, APP/PS1 mice did not differ in freezing levels during memory retrieval compared to WT controls. Unpaired $t$-test: $t_{13}$=0.52, p=0.61, WT (n=8), APP/PS1 (n=7). (**c**) At 20 weeks of age, APP/PS1 mice showed reduced freezing levels during memory retrieval compared to WT controls. Unpaired $t$-test: $t_{15}$=3.88, *p=0.002, WT (n=8), APP/PS1 (n=9). Graphs show mean ± s.e.m.

The online version of this article includes the following source data and figure supplement(s) for figure 1:

**Source data 1.** Individual datapoints of freezing levels in *Figure 1*.

**Figure supplement 1.** Amyloid beta plaque load in medial prefrontal cortex (mPFC) at 16 weeks and 20 weeks of age does not differ.

**Figure supplement 1—source data 1.** Individual datapoints of amyloid plaque in *Figure 1—figure supplement 1*.

## Results

### Progressive remote memory decline in APP/PS1 mice

We first determined whether remote memory is affected in APP/PS1 mice using contextual fear conditioning (CFC). For this, we subjected 12-week-old APP/PS1 mice and their WT littermates to CFC and 4 weeks later, at an age of 16 weeks, remote memory was assessed by re-exposing the mice to the FC context. Freezing levels of APP/PS1 mice did not differ compared with WT controls (*Figure 1a and b*). However, when APP/PS1 mice were trained at 16 weeks of age and tested at 20 weeks of age, we observed reduced freezing compared to WT littermates (*Figure 1a and c*). Of note, amyloid beta plaque size and number in the mPFC of APP/PS1 mice did not differ between 16 weeks and 20 weeks of age (*Figure 1*, *Figure 1—figure supplement 1*). Hence, remote memory progressively declines in APP/PS1 mice, but this is not mirrored by accumulating plaque load in the mPFC.

### APP/PS1 mice show an age-dependent increase in PV interneuron excitability

To determine whether changes in remote memory correlate with alterations in neuronal excitability, whole-cell patch-clamp recordings were performed in acute brain slices containing the mPFC (*Figure 2a*). To visualize PV neurons, APP/PS1 mice were crossed with PV-Cre transgenic mice to generate APP/PS1 PV-Cre mice, which were subsequently crossed with R26Al14 mice to create APP/PS1 PV-Cre-tdTomato and PV-Cre-tdTomato (control) mice. Recordings were made from PV interneurons and pyramidal neurons in the prelimbic subregion of the mPFC. At 16 weeks of age, PV cell resting membrane potential, rheobase, and action potential frequency in response to increasing current injections were unaltered in APP/PS1 PV-Cre-tdTomato mice compared to controls (*Figure 2b–e*). We detected only a decrease in membrane capacitance in PV cells of APP/PS1 mice (*Supplementary file 1, table 1A*). At this age, pyramidal cells in the mPFC of APP/PS1 mice also showed no alterations in resting membrane potential and action potential frequency (*Figure 2f–i*). However, pyramidal cells exhibited a decrease in rheobase (*Figure 2f*), as well as an increased action potential half-width (*Supplementary file 1, table 1B*).

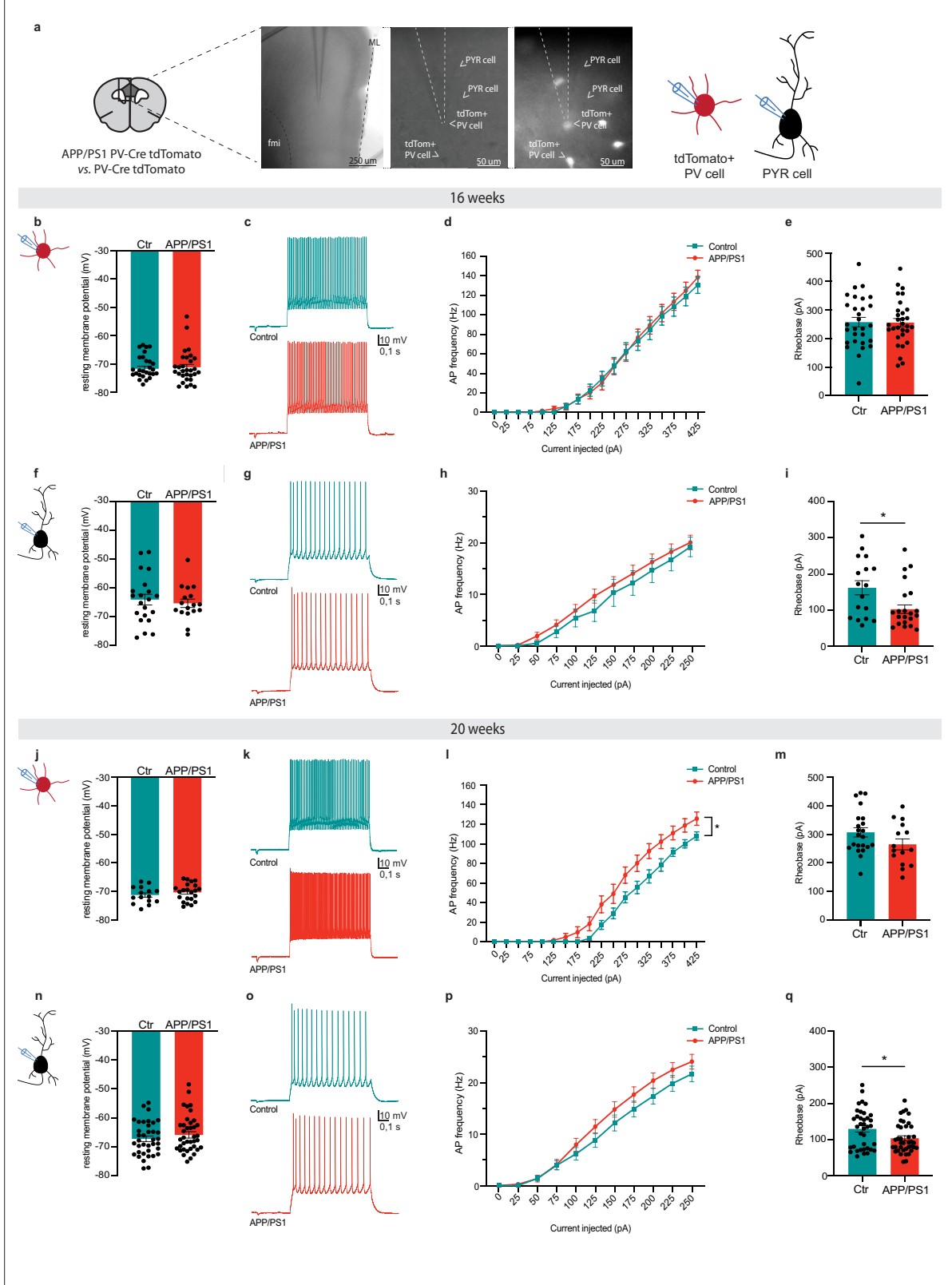

**Figure 2.** APP/PS1 mice show an age-dependent increase in parvalbumin (PV) interneuron excitability. (**a**) Schematic coronal brain section indicating the medial prefrontal cortex (mPFC) prelimbic region in dark gray, where tdTomato⁺ PV cells and pyramidal (PYR) cells were recorded in APP/PS1 PV-Cre tdTomato (APP/PS1) and PV-Cre tdTomato (control) mice. Representative differential interference contrast images (left) and a corresponding fluorescent image (right). fmi = forceps minor of the corpus callosum. ML = midline. Recordings were performed in 16- (**b–i**) and 20- (**j–q**) week-old mice. (**b**) Resting

*Figure 2 continued on next page*

*Figure 2 continued*

membrane potential was unaltered in PV cells at 16 weeks of age. Mann-Whitney test: $U$=433, p=0.98, n=29/30 cells, N=4 mice/genotype. Ctr = control. (**c**) Action potential (AP) firing of PV cells upon a depolarizing current step (250 pA). (**d**) AP frequency in PV cells in response to 0–425 pA depolarizing current steps did not differ between genotypes. Two-way repeated-measures ANOVA *genotype × current* $F_{(10,390)}$ = 0.44, p=0.93, n=29/30 cells, N=4 mice/genotype. (**e**) Rheobase was unchanged in PV cells. Unpaired *t*-test: $t_{57}$=0.08, p=0.93, n=29/30 cells, N=4 mice/genotype. Ctr = control. (**f**) Resting membrane potential of PYR cells did not differ between genotypes. Unpaired *t*-test: $t_{36}$=0.51, p=0.61, n=17/21 cells from N=4/6 control vs. APP/PS1 mice, respectively. Ctr = control. (**g**) AP firing of PYR cells upon a depolarizing current step (250 pA). (**h**) AP frequency in PYR cells in response to 0–250 pA depolarizing current steps did not differ between genotypes. Two-way repeated-measures ANOVA *genotype × current* $F_{(10,370)}$ = 0.25, p=0.99, n=17/21 from N=4/6 control vs. APP/PS1 mice, respectively. (**i**) APP/PS1 mice show a lower rheobase in PYR cells. Mann-Whitney test: $U$=111, *p=0.048, n=17/21 cells from N=4/6 control vs. APP/PS1 mice, respectively. Ctr = control. (**j**) Resting membrane potential was unaltered in PV cells at 20 weeks of age. Unpaired *t*-test: $t_{35}$=0.93, p=0.36, n=22/15 cells from N=6/7 control and APP/PS1 mice, respectively. Ctr = control. (**k**) AP firing of PV cells upon a depolarizing current step (250 pA). (**l**) APP/PS1 mice show an increased AP frequency in PV cells in response to 0–425 pA depolarizing current steps. Two-way repeated-measures ANOVA *genotype × current* $F_{(17,595)}$ = 4.05, *p<0.0001, n=22/15 cells, N=6/7 control and APP/PS1 mice, respectively. (**m**) Rheobase was unchanged in PV cells. Unpaired *t*-test: $t_{35}$=1.67, p=0.10, 22/15 cells, N=6/7 control and APP/PS1 mice, respectively. Ctr = control. (**n**) Resting membrane potential of PYR cells did not differ between genotypes. Unpaired *t*-test: $t_{74}$=0.92, p=0.36, n=37/39 cells, N=9 mice/genotype. Ctr = control. (**o**) AP firing of PYR cells upon a depolarizing current step (250 pA). (**p**) AP frequency in PYR cells in response to 0–250 pA depolarizing current steps did not differ between genotypes. Two-way repeated-measures ANOVA *genotype × current* $F_{(10,740)}$ = 1.80, p=0.08, n=37/39 cells, N=9 mice/genotype. (**q**) APP/PS1 mice show a decrease in PYR cell rheobase. Unpaired *t*-test: $t_{74}$=2.34, *p=0.022, n=37/39 cells, N=9 mice/genotype. Ctr = control. Graphs show mean ± s.e.m.

The online version of this article includes the following source data and figure supplement(s) for figure 2:

**Source data 1.** Individual datapoints of RMP, rheobase, and input-output curve in *Figure 2*.

**Figure supplement 1.** Somatostatin (SST) cell excitability is unaltered in the medial prefrontal cortex (mPFC) of 20-week-old APP/PS1 mice.

**Figure supplement 1—source data 1.** Individual datapoints of RMP, rheobase, and input-output curve in *Figure 2—figure supplement 1*.

Contrary to 16 weeks of age, action potential frequency in PV cells was enhanced in response to increasing current injections (*Figure 2j–m*) in APP/PS1 mice compared to WT controls at 20 weeks of age. No changes were observed in resting membrane potential and rheobase (*Figure 2n–q*, *Supplementary file 1, table 1C*) at this age. Similar to 16-week-old mice, pyramidal cells in 20-week-old APP/PS1 mice showed a decrease in rheobase, but no alterations in resting membrane potential and action potential frequency (*Figure 2n–q*). Action potential half-width of pyramidal cells was not altered in 20-week-old APP/PS1 mice (*Supplementary file 1, table 1D*).

Finally, we aimed to determine whether alterations in excitability were specific to PV interneurons, or if other memory-relevant interneurons, such as SST interneurons, were similarly affected. To do this, APP/PS1 mice were crossed with SST-Cre mice to generate APP/PS1-SST-Cre mice. Following micro-injection of AAV-hSyn-DIO-mCherry into the mPFC, recordings were obtained from SST neurons. At 20 weeks of age, SST interneurons showed no changes in excitability or membrane properties (*Figure 2*, *Figure 2—figure supplement 1*; *Supplementary file 1, table 1E*). Thus, although mPFC PV and pyramidal cells are both affected in APP/PS1 mice, only the development of PV cell hyperexcitability, and not of SST or pyramidal cells, mirrors the progressive decline in remote memory retrieval.

## Size and reactivation of the mPFC engram ensemble are unaffected in APP/PS1 mice

We previously demonstrated that reactivation of mPFC engram cells is required for remote memory retrieval 1 month after CFC using a viral-TRAP (targeted recombination in active populations) engram tagging approach (*Matos et al., 2019*). Here, we used the same technique to determine whether remote memory deficits in APP/PS1 mice at 20 weeks of age are reflected by alterations in properties and reactivation of the mPFC engram ensemble. Viral-TRAP allowed 4-hydroxytamoxifen (4TM)-controlled permanent expression of mCherry in activated neurons based on activation of the *Fos* promoter (*Cruz et al., 2013*; *Guenthner et al., 2013*; *Figure 3a*). Following viral-TRAP injection into the mPFC, APP/PS1 mice and their WT control littermates were subjected to CFC at either 12 weeks or 16 weeks of age, after which CFC-activated cells were permanently labeled with mCherry upon 4TM injection (*Figure 3—figure supplement 1*). Four weeks later, at either 16 weeks or 20 weeks of age, remote memory retrieval was assessed (*Figure 3b*). We confirmed that APP/PS1 mice have a remote memory deficit at 20 weeks, but not at 16 weeks, of age (*Figure 3—figure supplement 2*).

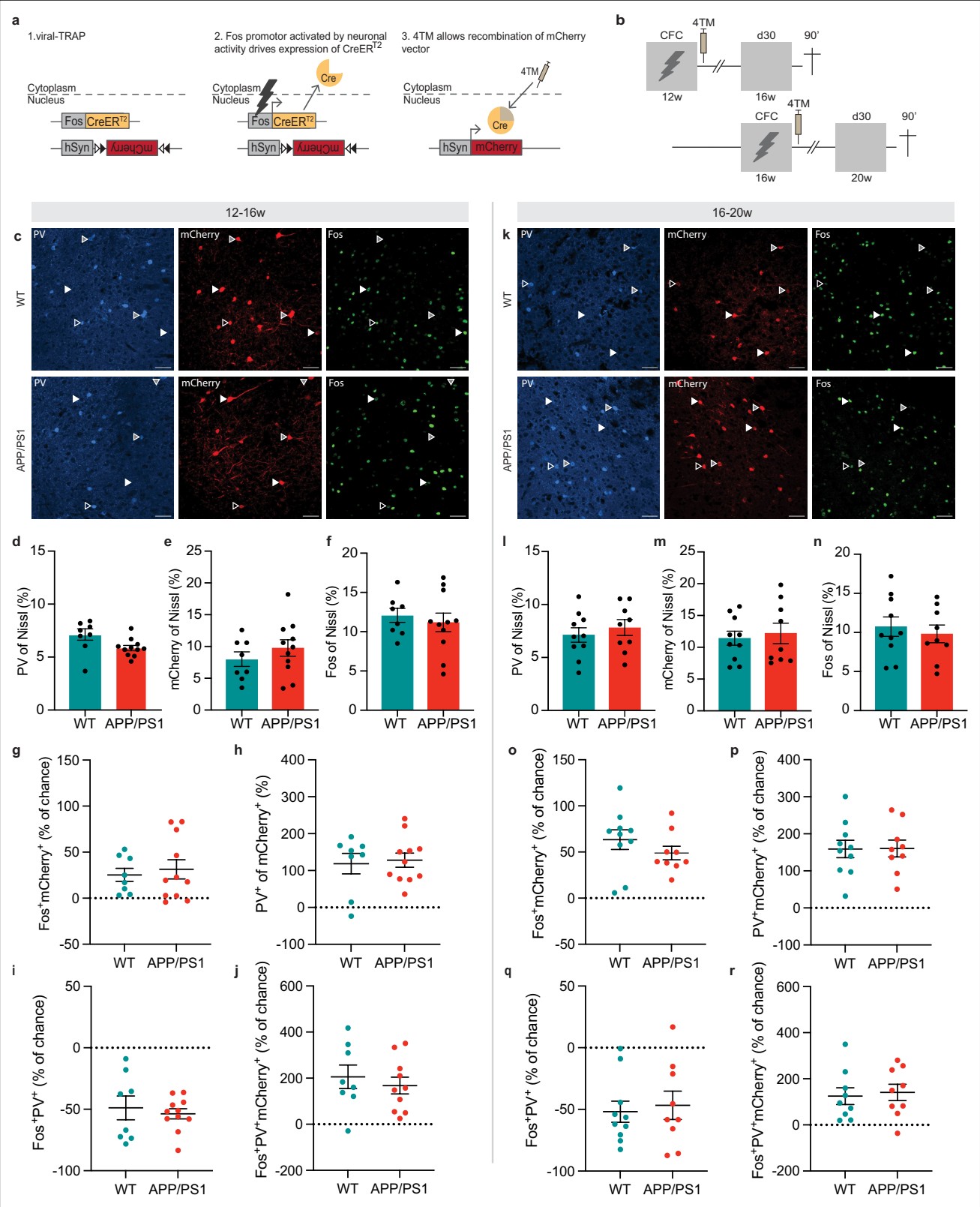

**Figure 3.** Size and reactivation of the medial prefrontal cortex (mPFC) engram ensemble, as well as parvalbumin (PV) interneuron (re)activation, are unaffected in APP/PS1 mice. (**a**) Schematic representation of the viral-TRAP method. A viral cocktail of AAV-Fos-CreER[T2] and Cre-dependent AAV-hSyn-DIO-mCherry was injected into the mPFC, allowing irreversible expression of mCherry upon neuronal activity and systemic injection of 4-hydroxytamoxifen (4TM). (**b**) Schematic timeline depicting contextual fear conditioning (CFC) and engram tagging at 12 weeks (WT n=8, APP/PS1

*Figure 3 continued on next page*

*Figure 3 continued*

n=11 mice) or 16 weeks (WT n=10, APP/PS1 n=9 mice) old. On day 30 after training, mice underwent a remote memory test at 16 weeks or 20 weeks of age and were perfused 90 min later. (**c**) Representative images at 12–16 weeks showing PV⁺, mCherry⁺, and Fos⁺ cells in WT (top row) and APP/PS1 (bottom row) mice. Nissl staining (general neuronal marker) is not shown. White arrowheads indicate reactivated neurons (Fos⁺/mCherry⁺ cells). Gray arrowheads indicate PV cells that are part of the engram (PV⁺/mCherry⁺ cells). Empty arrowheads indicate reactivated PV neurons (Fos⁺/mCherry⁺/PV⁺ cells). Scale bar = 50 µm. Colocalization of cell-type markers is shown for 16- (**d–j**) and 20- (**l–r**) week-old mice. (**d**) Percentage of PV cells did not differ between genotypes. Unpaired *t*-test: $t_{17}$=2.04, p=0.06. (**e**) Percentage of mCherry⁺ cells did not differ between genotypes. Unpaired *t*-test: $t_{17}$=1.02, p=0.32. (**f**) Percentage of Fos⁺ cells did not differ between genotypes. Unpaired *t*-test: $t_{17}$=0.53, p=0.60. (**g**) In both genotypes, Fos and mCherry overlapped above chance level. One-sample *t*-test: WT $t_7$=3.59, *p=0.009; APP/PS1 $t_{10}$=3.02, *p=0.013. (**h**) Both genotypes showed above chance overlap of PV and mCherry. One-sample *t*-test: WT $t_7$=4.27, *p=0.004; APP/PS1 $t_{10}$=6.69, *p<0.0001. (**i**) Both genotypes showed below chance colocalization of Fos and PV. One-sample *t*-test: WT $t_7$=5.06, *p=0.002; APP/PS1 $t_{10}$=12.99, *p<0.0001. (**j**) Fos, PV, and mCherry⁺ overlap did not differ from chance level in both genotypes. One-sample *t*-test: WT $t_7$=4.05, *p=0.005; APP/PS1 $t_9$=4.60, *p=0.001. (**k**) Representative images at 16–20 weeks showing PV⁺, mCherry⁺, and Fos⁺ cells in WT (top row) and APP/PS1 (bottom row) mice. (**l**) Percentage of PV cells did not differ between genotypes. Unpaired *t*-test: $t_{17}$=0.67, p=0.51. (**m**) Percentage of mCherry⁺ cells did not differ between genotypes. Unpaired *t*-test: $t_{17}$ = 0.42, p=0.68. (**n**) Percentage of Fos⁺ cells did not differ between genotypes. Unpaired *t*-test: $t_{17}$=0.55, p=0.59. (**o**) In both genotypes, Fos and mCherry colocalized above chance level. One-sample *t*-test: WT $t_9$=5.97, *p=0.0002; APP/PS1 $t_8$=6.63, *p=0.0002. (**p**) Both genotypes showed above chance overlap of PV and mCherry. One-sample *t*-test: WT $t_9$=6.73, *p<0.0001; APP/PS1 $t_8$=7.05, *p=0.0001. (**q**) Both genotypes showed below chance colocalization of Fos and PV. One-sample *t*-test: WT $t_9$=6.04, *p=0.002; APP/PS1 $t_8$=4.07, *p=0.004. (r) Fos, PV, and mCherry overlap did not differ from chance level in both genotypes. One-sample *t*-test: WT $t_8$=3.45, *p=0.009; APP/PS1 $t_8$=4.00, *p=0.004. Graphs show mean ± s.e.m.

The online version of this article includes the following source data and figure supplement(s) for figure 3:

**Source data 1.** Individual datapoints of immunohistochemistry counts in *Figure 3*.

**Figure supplement 1.** Representative example of mCherry expression in the medial prefrontal cortex (mPFC) after contextual fear conditioning (CFC) and 4-hydroxytamoxifen (4TM) treatment in APP/PS1 mice and WT controls, at 12–16 weeks and 16–20 weeks.

**Figure supplement 2.** APP/PS1 mice show remote memory impairment at 20, but not 16, weeks of age.

**Figure supplement 2—source data 1.** Individual datapoints of freezing levels in *Figure 3—figure supplement 2*.

**Figure supplement 3.** Size and reactivation of the medial prefrontal cortex (mPFC) engram ensemble, as well as parvalbumin (PV) interneuron (re) activation, are unaffected in APP/PS1 mice.

**Figure supplement 3—source data 1.** Individual datapoints of immunohistochemistry counts in *Figure 3—figure supplement 3*.

**Figure supplement 4.** Increased reactivation of engram cells in 20- vs.16-week-old WT and APP/PS1 mice.

**Figure supplement 4—source data 1.** Individual datapoints of immunohistochemistry counts in *Figure 3—figure supplement 4*.

Mice were perfused 90 min after the retrieval test, followed by immunohistochemical staining of mPFC sections for Nissl (general neuronal marker), PV, and Fos (representing neurons activated during memory retrieval; *Figure 3c and k*). The percentage of PV⁺ (*Figure 3d and l*), mCherry⁺ (*Figure 3e and m*), and Fos⁺ (*Figure 3f and n*) neurons did not differ between control and APP/PS1 mice at either 16 weeks or 20 weeks of age.

Reactivation of mCherry⁺ engram cells was assessed by examining colocalization of Fos⁺ and mCherry⁺ cells and comparing between genotypes. In line with our previous findings (*Matos et al., 2019*), Fos and mCherry colocalized above chance level in both control and APP/PS1 mice at 16 weeks of age, pointing to preferential reactivation of engram cells when remote memory retrieval is still intact (*Figure 3g*). Accordingly, colocalization of Fos with mCherry⁺ cells (mCherry⁺/Nissl⁺) was enhanced compared to colocalization with mCherry⁻ cells (mCherry⁻/Nissl⁺) (*Figure 3—figure supplement 3a*), in both genotypes. Surprisingly, we observed similar preferential colocalization of Fos⁺ and mCherry⁺ cells in both genotypes at 20 weeks of age, but no genotype difference (*Figure 3o*, *Figure 3—figure supplement 3e*). Finally, we compared reactivation levels between the 12–16 weeks group and the 16–20 weeks group (*Figure 3—figure supplement 4a*). A two-way ANOVA revealed a main effect of age, but not genotype, suggesting that reactivation of tagged neurons increased with age and regardless of genotype. Thus, while APP/PS1 mice showed impaired remote memory expression when they were 20 weeks of age, reactivation of the engram ensemble in the mPFC seemed unaffected.

## PV interneuron (re)activation is unaffected in APP/PS1 mice

Next, we quantified colocalization of PV⁺ and mCherry⁺ cells, representing the number of PV cells initially recruited to the mPFC engram ensemble. In both age groups and genotypes, PV⁺ cells and mCherry⁺ cells colocalized above chance level, suggesting that PV neurons were preferentially recruited to the mPFC engram ensemble (*Figure 3h and p*). No differences were found between

APP/PS1 mice and WT mice. Similarly, colocalization of PV$^+$ with mCherry$^+$ (mCherry$^+$/Nissl$^+$) cells was enhanced compared to the mCherry$^-$ (mCherry$^-$/Nissl$^+$) population in both genotypes and age groups (*Figure 3—figure supplement 3b and f*). A comparison between age groups revealed no differences (*Figure 3—figure supplement 4b*). Furthermore, we studied the overlap between Fos$^+$ and PV$^+$/mCherry$^-$ cells, reflecting the number of activated non-tagged PV cells during memory retrieval. We did not find differences between genotypes at 16 weeks and 20 weeks of age (*Figure 3—figure supplement 3c and g*), but in all groups, PV$^+$ and Fos$^+$ cells colocalized below chance level (*Figure 3i and q*). Likewise, Fos$^+$ cells showed decreased colocalization with the PV$^+$/mCherry$^-$ (PV$^+$/mCherry$^-$/Nissl$^+$) population compared with the PV$^-$/mCherry$^-$ (PV$^-$/mCherry$^-$/Nissl$^+$) population during memory retrieval. Again, no differences were found between age groups (*Figure 3—figure supplement 4c*).

Finally, we quantified the colocalization of PV$^+$, Fos$^+$, and mCherry$^+$ cells, representing PV cells that were tagged during CFC and reactivated during remote memory expression. Strikingly, the overlap between PV$^+$, Fos$^+$, and mCherry$^+$ was above chance level in APP/PS1 mice and control mice at both ages, but this again did not differ between genotypes (*Figure 3j and r*). Similarly, the percentage of Fos$^+$ cells was higher in the PV$^+$/mCherry$^+$ population than the PV$^+$/mCherry$^-$ population in APP/PS1 and control mice at both ages (*Figure 3—figure supplement 3d and h*). This indicates that tagged PV$^+$ cells were preferentially reactivated during remote memory expression, in contrast to the reduced activation of (non-tagged) PV$^+$ cells that were not part of the engram population. However, this did not differ between genotypes in 16- and 20-week-old mice, nor did it differ between the 12–16 weeks and 16–20 weeks age groups (*Figure 3—figure supplement 4d*). Thus, while PV cells are hyperexcitable at 20 weeks of age in APP/PS1 mice, they do not seem to be differently recruited during memory encoding nor (re)activated during remote memory retrieval.

## Perisomatic PV labeling is increased on engram cells in 20-week-old APP/PS1 mice

Neocortical PV interneurons predominantly synapse onto the soma, proximal dendrites, and axon initial segment of other neurons to control their output (*Tremblay et al., 2016*). Therefore, we hypothesized that differential PV innervation of engram vs. non-engram cells might underly the remote memory deficit in 20-week-old APP/PS1 mice. As a first step to address this, we measured the surface area of PV$^+$ labeling in close proximity to the soma of mCherry$^+$/PV$^-$ cells and compared this to neighboring mCherry$^-$/PV$^-$ cells (*Figure 4*; *Trouche et al., 2013*). In the 12- to 16-week groups, we found a similar amount of PV labeling around mCherry$^+$ cells compared to mCherry$^-$ cells, for both control and APP/PS1 mice (*Figure 4a–c*). In contrast, PV labeling around mCherry$^+$ cells was enhanced compared to mCherry$^-$ cells in APP/PS1, but not in control mice in the 16- to 20-week groups (*Figure 4d–f*), suggesting that inhibitory input may be selectively enhanced in mPFC engram cells of 20-week-old APP/PS1 mice.

## Engram cells of 20-week-old APP/PS1 mice receive increased inhibitory input

We next aimed to confirm that inhibitory synaptic transmission is altered in engram cells of APP/PS1 mice using whole-cell patch-clamp electrophysiology. Viral-TRAP was used to permanently label CFC-activated engram cells in the mPFC, and 4 weeks later, at 20 weeks of age, we subjected mice to a memory retrieval session, after which we immediately generated acute brain slices to measure spontaneous inhibitory and excitatory postsynaptic currents (sIPSCs and sEPSCs, respectively) from mCherry$^+$, and neighboring mCherry$^-$, pyramidal cells (*Figure 5a–d*). A two-way repeated-measures ANOVA revealed an interaction effect between genotype and cell type for sIPSC frequency. Post hoc analysis confirmed that in APP/PS1 mice, sIPSC frequency was increased in mCherry$^+$ cells compared to neighboring mCherry$^-$ cells, which was not observed in WT control mice (post hoc Bonferroni test: APP/PS1 mCherry$^+$ vs. mCherry$^-$ p=0.011) (*Figure 5e*). The sIPSC amplitude did not differ between mCherry$^+$ and mCherry$^-$ cells in both APP/PS1 and WT mice (*Figure 5f*). Finally, we measured sEPSC frequency and amplitude onto mCherry$^+$ and mCherry$^-$ cells. For sEPSC frequency, a two-way repeated-measures ANOVA did not detect an interaction effect but revealed a main effect of cell population, without a significant post hoc difference, indicating that spontaneous excitatory input onto engram cells was enhanced independent of genotype (*Figure 5g*). Similar to sIPSC amplitude, there was no difference in sEPSC amplitude between mCherry$^+$ and mCherry$^-$ cells (*Figure 5h*). Hence, excitatory

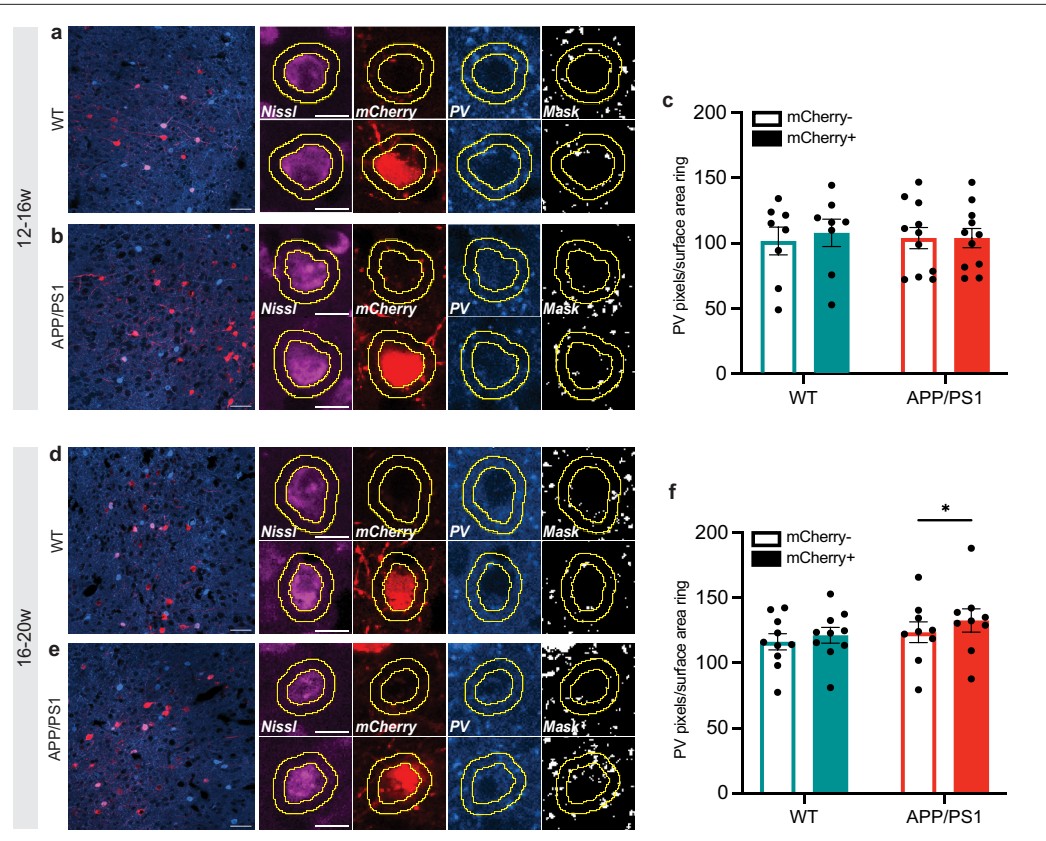

**Figure 4.** Perisomatic parvalbumin (PV) labeling is increased on engram cells in 20-week-old APP/PS1 mice. (a–b) Left: Representative image of WT (a) and APP/PS1 (b) mice from the 12- to 16-week groups showing PV staining and mCherry$^+$ cells in the mPFC. Scale bar = 50 μm. Right: Examples of an mCherry$^-$ (top row) and mCherry$^+$ (bottom row) cells. Soma outline is based on Nissl. PV signal was masked and measured inside the ring surrounding the soma. Scale bar = 10 μm. (c) PV labeling around mCherry$^+$ cells did not differ from mCherry$^-$ cells in WT and APP/PS1 mice. Two-way repeated-measures ANOVA *cell population:* $F_{(1,17)}$ = 2.25; p=0.15; WT (n=8), APP/PS1 (n=11). (d) Left: Representative image of WT (d) and APP/PS1 (e) mice from the 16- to 20-week groups showing PV staining and mCherry$^+$ cells in the mPFC. Scale bar = 50 μm. Right: Examples of an mCherry$^-$ (top row) and mCherry$^+$ (bottom row) cell. Soma outline is based on Nissl. PV signal was masked and measured inside the ring surrounding the soma. Scale bar = 10 μm. (f) An increased amount of PV labeling around mCherry$^+$ cells was found compared to mCherry$^-$ cells in APP/PS1 mice but not control mice. Two-way repeated-measures ANOVA *cell population*: F(1,17) = 21.74; *p=0.0002. Post hoc Bonferroni test: APP/PS1 mCherry$^+$ vs. mCherry$^-$ *p=0.0015; WT (n=10), APP/PS1 (n=9). Graphs show mean ± s.e.m.

The online version of this article includes the following source data for figure 4:

**Source data 1.** Individual datapoints of perisomatic parvalbumin (PV) labeling in *Figure 4*.

drive onto mPFC engram cells is modestly enhanced in control and APP/PS1 mice, whereas inhibitory input onto the same cells is selectively augmented in APP/PS1 mice.

## Discussion

In this study, we demonstrate an age-dependent decline in remote memory retrieval in APP/PS1 mice, which coincides with progressive PV interneuron hyperexcitability in the mPFC. Strikingly, this remote memory deficit does not seem to be mirrored by alterations in initial activation of neurons during learning, reactivation of the engram ensemble during memory retrieval, nor (re)activation of PV cells in the mPFC, as assessed using Fos expression. Interestingly, the amount of PV labeling on engram cells was increased compared to non-engram cells in APP/PS1 mice, which was not observed in control mice. In addition, we observed enhanced spontaneous inhibitory input onto engram cells in APP/PS1 but not control mice. Together, these findings suggest that increased innervation of mPFC engram

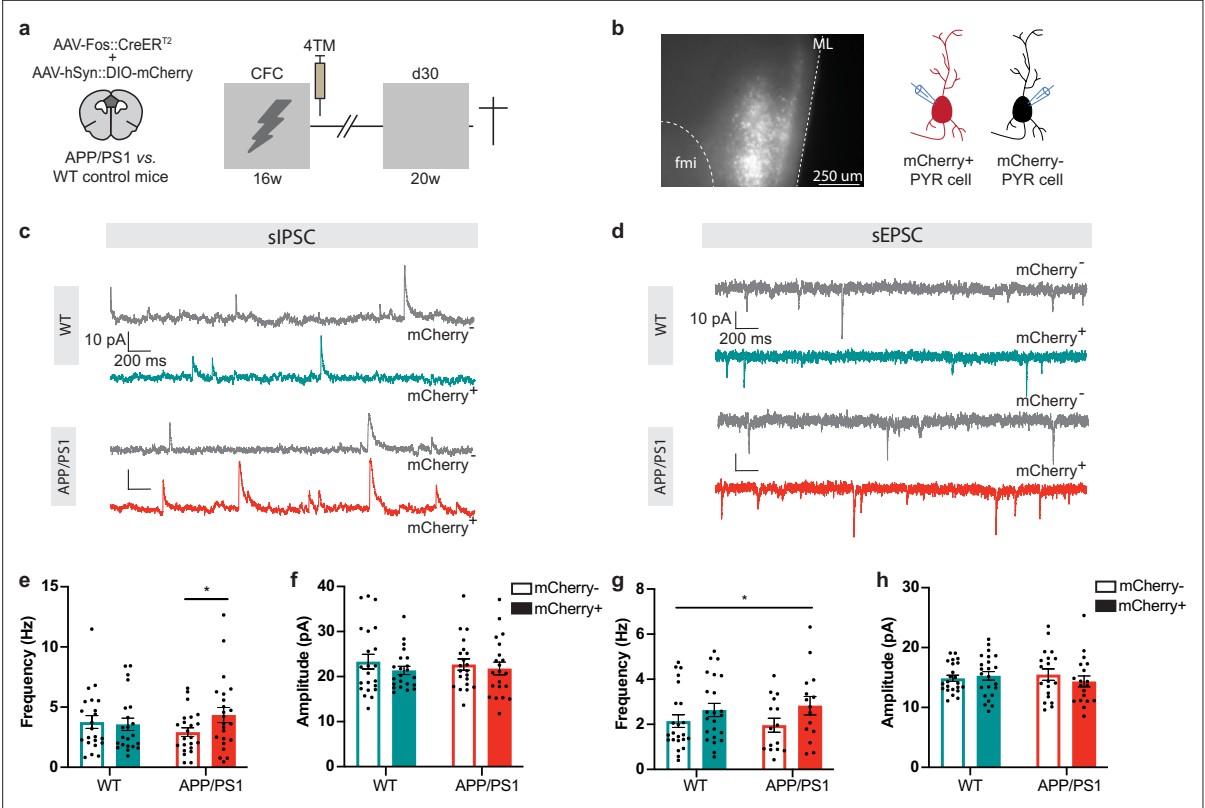

**Figure 5.** Engram cells of 20-week-old APP/PS1 mice receive increased inhibitory input. (**a**) Coronal brain section indicating the medial prefrontal cortex (mPFC) region (dark gray) where AAV-Fos-CreER[T2] and Cre-dependent AAV-hSyn-DIO-mCherry were injected. Mice underwent contextual fear conditioning (CFC) at 16 weeks of age, and engram cells were tagged. Thirty days after CFC, mice were re-exposed to the training context and then immediately sacrificed for whole-cell patch-clamp electrophysiology. (**b**) Left: Representative image showing labeled mCherry[+] engram cells in the mPFC. Right: Recordings were made from mCherry[+] and mCherry[-] pyramidal cells. fmi = forceps minor of the corpus callosum. ML = midline. (**c**) Example spontaneous inhibitory postsynaptic current (sIPSC) traces of mCherry[+] and mCherry[-] for WT control and APP/PS1 mice. (**d**) Example spontaneous excitatory postsynaptic current (sEPSC) traces of mCherry[+] and mCherry[-] cells for WT and APP/PS1 mice. (**e**) Frequency of sIPSCs differed between mCherry[+] and mCherry[-] cells in APP/PS1, but not WT, mice. Two-way repeated-measures ANOVA *genotype × cell type* $F_{(1,43)}$ = 5.44, *p=0.024. Post hoc Bonferroni APP/PS1 mCherry[+] vs. mCherry[-] *p=0.011. n=22 per cell type from N=6 WT mice, n=23 per cell type from N=7 APP/PS1 mice. (**f**) sIPSC amplitude did not differ between cell type and genotype. Two-way repeated measure ANOVA *cell-type* $F_{(1,43)}$ = 1.24, *p* = 0.27. Two-way repeated measure ANOVA *genotype* $F_{(1,43)}$ = 0.01, *p* = 0.93. *n* = 22 per cell-type from N = 6 WT mice, *n* = 23 per cell-type from N = 7 APP/PS1 mice(**g**) Frequency of sEPSCs was enhanced in mCherry[+] cells compared to mCherry[-] cells in both genotypes. Two-way repeated-measures ANOVA *cell type* $F_{(1,36)}$ = 7.26, *p=0.011, n=20 per cell type from N=6 WT mice, n=23 per cell type from N=7 APP/PS1 mice. (**h**) sEPSC amplitude did not differ between cell type and genotype. Two-way repeated measure ANOVA *cell-type* $F_{(1,40)}$ = 0.32, *p* = 0.57. Two-way repeated measure ANOVA *genotype* $F_{(1,40)}$ = 0.03, *p* = 0.86. *n* = 20 per cell type from N = 6 WT mice, *n* = 23 per cell-type from N = 7 APP/PS1 mice. Graphs show mean ± s.e.m.

The online version of this article includes the following source data for figure 5:

**Source data 1.** Individual datapoints of spontaneous inhibitory postsynaptic current (sIPSC) and spontaneous excitatory postsynaptic current (sEPSC) measurements in *Figure 5*.

cells by hyperexcitable PV interneurons, rather than alterations in engram composition, is responsible for remote memory deficits in APP/PS1 mice.

APP/PS1 mice recapitulate pathological changes in AD, including amyloid beta plaque deposition, astrogliosis, and microgliosis (*Jackson et al., 2013*; *Liu et al., 2020*). Here, we show that this mouse model also shows an age-dependent impairment in remote memory, mirroring the gradual loss of remote memories in AD patients. These patients initially experience anterograde and temporally graded retrograde amnesia, which progressively develops into a more profound retrograde memory impairment (*El Haj et al., 2015*; *Knopman et al., 2021*). In line with this, we have repeatedly reported that APP/PS1 mice show recent memory deficits already at 12 weeks of age (*Kater et al., 2023*; *Végh et al., 2014*). Here, we show that when mice are trained at 12 weeks of age and memory is assessed at a remote time point, i.e., 4 weeks later, remote memory retrieval is still unaffected. However, remote

memory is impaired when APP/PS1 mice are trained at 16 weeks of age and memory retrieval is evoked 1 month later. This suggests that a remote memory can be formed even when recent memory expression is already compromised, indicating that the remote memory deficit in 20-week-old APP/PS1 mice is not a continuation of earlier recent memory impairments. Hence, whereas initially only recent memory is affected in APP/PS1 mice, i.e., mimicking early temporally graded retrograde amnesia in patients, remote memory can no longer be retrieved at a later age, resembling the more severe retrograde amnesia, as observed in late-stage AD patients.

To our knowledge, remote CFC memory has not been previously studied in APP/PS1 mice before. Contrary to our results, however, 5xFAD mice show a remote memory deficit (30 days after CFC) prior to recent memory deficit (1 day after CFC) before 4 months of age (*Kimura and Ohno, 2009*). Given the slower development of remote memory impairment compared with recent memory loss in AD patients, APP/PS1 mice may serve as a more suitable model for investigating memory decline in AD. Although extrapolations to human AD should be made with caution, the distinction between recent and remote memory mechanisms in mice could provide useful information for targeting early- and late-stage memory decline in humans more specifically.

At 20 weeks of age, APP/PS1 mice exhibit PV cell hyperexcitability in the mPFC, which was not present in 16-week-old animals, thereby mirroring the development of remote memory dysfunction. Although the excitability measurements were performed in APP/PS1-PV-Cre-tdTomato mice, and not in the APP/PS1 parental line, we previously found that these transgenic mouse lines exhibit comparable amyloid pathology (both soluble and insoluble amyloid beta levels), as well as similar spatial memory deficits (*Hijazi et al., 2020a*; *Kater et al., 2023*). Thus, our observations indicate that the APP/PS1 PV-Cre-tdTomato and APP/PS1 lines are similar in terms of pathology and behavior. Nonetheless, further work is needed to identify a causal link between PV cell hyperexcitability and remote memory impairment. The hyperexcitability is specific to PV-expressing interneurons, as the same change was not observed in SST cells. Differences in intrinsic excitability are known to affect the probability that a neuron is recruited to an engram (*Yiu et al., 2014*; *Zhou et al., 2009*), and PV interneurons control the size of an engram (*Morrison et al., 2016*). However, we found that both the size and PV cell content of the mPFC engram ensemble did not differ between APP/PS1 and control mice, suggesting that memory allocation is not affected. Similarly, memory allocation to neurons in the hippocampal dentate gyrus in APP/PS1 mice is not altered even though recent memory retrieval is impaired (*Perusini et al., 2017*; *Roy et al., 2016*). We speculate that in 12-week-old APP/PS1 mice, memory allocation to hippocampal and cortical engram cells is sufficient to promote consolidation of the memory into cortical engram circuits, resulting in the formation of a remote memory, independent of a recent memory retrieval deficit. This points to a recent and remote memory retrieval problem in APP/PS1 mice, as opposed to an encoding deficit. However, in contrast to reduced DG engram reactivation in APP/PS1 mice during recent memory expression (*Perusini et al., 2017*; *Roy et al., 2016*), reactivation of the mPFC engram during remote retrieval at 20 weeks did not seem to differ between APP/PS1 and control mice in our study, suggesting that the remote memory deficit is caused by a different mechanism.

Surprisingly, at 20 weeks of age, the percentage of activated PV cells during memory retrieval was not affected in APP/PS1 mice, despite the overall PV cell hyperexcitability in the mPFC. However, in both control and APP/PS1 mice, and independent of age, PV cells appear to be inhibited during remote retrieval, as activation of the non-tagged PV population was below chance level. In line with our data, fear expression (i.e. freezing behavior) is causally related to reduced PV interneuron activity in the mPFC (*Courtin et al., 2014*; *Cummings and Clem, 2020*). Interestingly, we also demonstrate that the small subset of tagged PV cells was reactivated above chance level during remote memory retrieval in both control and APP/PS1 mice. While interneurons exhibit learning-induced molecular, structural, and electrophysiological plasticity (*Donato et al., 2013*; *Trouche et al., 2013*; *Wolff et al., 2014*), it remains to be determined whether the subset of learning-tagged PV cells is functionally involved in remote memory expression. Furthermore, the opposing level of activity of learning-tagged and non-tagged PV interneurons during remote memory retrieval (i.e. enhanced vs. decreased activity, respectively) may both be relevant for memory expression. To determine the causal contribution of PV cell changes in APP/PS1 mice, it may thus be necessary to simultaneously manipulate both PV populations in opposite directions, i.e., stimulate non-tagged PV cells and inhibit learning-activated PV cells, or to selectively target one of these subsets. This requires use of an intersectional approach (*Fenno*

*et al., 2014*). However, as in our experiments, both engram tagging (viral-TRAP) and PV labeling (PV-Cre mice) depend on Cre-mediated recombination. This presents technical challenges and remains to be addressed with novel complementary techniques in the future.

Despite the lack of changes in (re)activation of PV interneurons, our data indicate that augmented inhibitory input onto engram cells in the mPFC may underlie the remote memory deficit in 20-week-old APP/PS1 mice. Interestingly, both WT and APP/PS1 mice showed an increase in sEPSC frequency onto engram cells, suggesting that increased excitatory input is a consequence of memory retrieval and not affected by genotype. However, only in APP/PS1 mice did the augmented excitatory input coincide with an elevation of inhibitory input onto engram cells. The resulting imbalance between excitation and inhibition could therefore potentially disrupt the precise control of engram reactivation and contribute to the observed remote memory impairment. The concurrent increase in PV labeling around engram cells and PV cell hyperexcitability at 20 weeks strongly suggests that the altered inhibitory drive is mediated by PV cells. Although PV$^+$ cells were not included in this analysis and we excluded non-neuronal cells based on the area of the Nissl stain, the mCherry$^-$ population was potentially more heterogenous than the mCherry$^+$ population, which may have contributed to the differences we observed. Furthermore, although SST cell excitability is not affected in 20-week-old APP/PS1 mice, we cannot exclude the possibility that input from SST cells or other inhibitory neurons contributes to the enhanced sIPSC frequency in mPFC engram cells.

Despite the difference in inhibitory input, overall reactivation of engram cells appears unchanged in APP/PS1 mice. This paradox, where altered synaptic input does not lead to a change in engram reactivation, suggests we may not have been able to detect changes in engram function using Fos expression as a proxy for neuronal activity. Fos is a widely used tool to study neuronal activity and engram ensembles (*Cruz et al., 2013*); however, Fos expression does not accurately capture changes in the level of neuronal activity, nor in synchronization of activity, the latter being a key characteristic of a functional neuronal ensemble (*Buzsáki, 2010*; *Josselyn and Tonegawa, 2020*; *Zhou et al., 2020*). Selectively enhanced activity of shock-responsive PV cells in the retrosplenial cortex has recently been observed during subsequent freezing epochs using in vivo calcium imaging, and this PV ensemble dynamic is disrupted in 5xFAD mice (*Park et al., 2024*). The reactivated subset of PV cells in our study may reflect the shock-responsive PV ensemble and potentially regulate pyramidal engram cell firing patterns. PV interneurons are critical for the rhythmic firing of pyramidal cells (*Buzsáki, 2002*; *Sohal et al., 2009*), which is essential for memory processes (*Courtin et al., 2014*; *Xia et al., 2017*). In APP/PS1 mice, hyperexcitability of PV cells may disrupt their synchronicity and thereby the synchronous firing of other engram cells during memory retrieval, which is not captured by Fos expression. Interestingly, we recently found a progressive disruption of oscillatory network synchrony between the mPFC and hippocampus in APP/PS1 PV-Cre mice (*van Heusden et al., 2023*). However, whether the observed PV cell hyperexcitability directly contributes to changes in inter-regional synchrony, and whether this leads to alterations at a network level, i.e., increased inhibitory input on engram cells, and consequently to memory deficits, remains to be elucidated in future studies.

Alternatively, changes in the output of mPFC engram cells might mediate the remote memory impairment. PV interneurons are known to exert inhibition at the soma and axonal initial segment of pyramidal cells and thereby suppress their output (*Somogyi et al., 1982*; *Veres et al., 2014*). Enhanced inhibition at these sites may interfere with memory retrieval independent of the dendritic excitatory inputs that induce Fos expression. PV neurons in the mPFC have also been shown to inhibit pyramidal neurons projecting to the basolateral amygdala, and disinhibition of these pyramidal cells evokes fear expression (*Courtin et al., 2014*). Hence, PV cell-mediated alterations in firing synchronicity and/or output of engram cells might underlie the observed remote memory deficit in APP/PS1 mice and are therefore important topics for future research. Additionally, long-range GABAergic projections of fast-spiking PV neurons in the mPFC have been identified (*Lee et al., 2014*) and may also contribute to remote memory impairment without altering local Fos expression.

Prior studies have shown that neurons in the vicinity of amyloid beta plaques show increased excitability (*Busche et al., 2008*). We demonstrated that PV neurons in the CA1 are hyperexcitable and that treatment with BACE1 inhibitors, i.e., reducing amyloid beta levels, rescues PV excitability (*Hijazi et al., 2020a*). In line with this, we also reported that the addition of amyloid beta to hippocampal slices increases PV excitability, without altering pyramidal cell excitability (*Hijazi et al., 2020a*). Finally, applying amyloid beta to an induced mouse model of PV hyperexcitability further impairs PV function

(*Hijazi et al., 2020b*). Since amyloid beta plaque load in the mPFC remains comparable between 16- and 20-week-old APP/PS1 mice, the observed increased excitability is unlikely the result of changes in insoluble amyloid beta levels. Previous data from our lab show that soluble amyloid beta is already present as early as 6 weeks of age and becomes more prominent at 24 weeks of age (*Kater et al., 2023*; *Végh et al., 2014*). The progressive increase in soluble amyloid beta levels may contribute to the emergence of PV cell hyperexcitability. We hypothesize that the hyperexcitability induced by amyloid beta may result from disrupted ion channel function, as PV neuron dysfunction can result from altered potassium (*Olah et al., 2022*) and sodium channel activity (*Verret et al., 2012*).

In addition to excitability changes in PV cells, we observed a decreased rheobase in mPFC pyramidal cells of APP/PS1 mice at both 16 weeks and 20 weeks of age. Interestingly, an age-related decrease in rheobase in pyramidal cells has been previously reported (*Popescu et al., 2021*) and is associated with spatial working memory deficits (*Moore et al., 2023*). This suggests that pyramidal cells in young adult APP/PS1 mice show characteristics that resemble the aged brain. Whether and how the altered rheobase in mPFC pyramidal cells contributes to dysfunctional remote memory in APP/PS1 mice remains to be elucidated. Moreover, it is relevant to investigate whether changes in PV and PYR cell excitability, as well as input onto engram cells in the mPFC, become more pronounced at later disease stages. Nonetheless, by focusing on early disease timepoints in the present study, we aimed to understand the initial circuit-level changes in AD and identify targets for early therapeutic intervention.

Notably, we observed a rightward shift in the AP frequency curve in response to increasing current injections in PV cells when comparing 16- and 20-week-old control mice, suggestive of an age-dependent change in PV interneuron excitability. In support of this, human studies have shown a weakening of action potentials in fast-spiking cortical interneurons with age, caused by an increase in spike half-width and a decrease in action potential rise speed (*Szegedi et al., 2024*). Furthermore, we also observed an increase in the amount of overlap between Fos[+] and mCherry[+] engram cells when comparing the 12- to 16-week and 16- to 20-week age groups. This finding should be interpreted with caution, as the experimental groups were processed in separate cohorts, with viral injections and 4TM-induced tagging performed at different moments in time. This may have contributed to the observed differences between ages. Nonetheless, it would be interesting to investigate whether these changes in PV cell excitability and engram reactivation continue with age and what the potential consequences might be at the engram level, in terms of memory storage and retrieval.

To conclude, we demonstrate that PV cell hyperexcitability and enhanced inhibitory input onto mPFC engram cells coincides with an age-dependent remote memory decline in an AD mouse model. Moving forward, it is important to focus on changes in network synchrony and output of cortical engram cells as potential contributors to remote memory impairment. Understanding these network dynamics and the specific cells that are affected in AD is crucial for the identification of new therapeutic targets for alleviation of memory loss in AD.

## Materials and methods

### Key resources table

| Reagent type (species) or resource | Designation | Source or reference | Identifiers | Additional information |
|---|---|---|---|---|
| Strain, strain background construct (*M. musculus*) | APP/PS1, APPswe,PSEN1de9 | The Jackson Laboratory | Stock number 004462, RRID:MMRRC_034829-JAX | Male |
| Strain, strain background (*M. musculus*) | PV-Cre, B6.129P2-Pvalbtm1(cre)Arbr/J | The Jackson Laboratory | Stock number 017320 RRID:IMSR_JAX:017320 | Male |
| Strain, strain background (*M. musculus*) | SST-Cre, Ssttm2.1(cre)Zjh/J | The Jackson Laboratory | Stock number 013044 RRID:IMSR_JAX:013044 | Male |
| Strain, strain background (*M. musculus*) | R26AI14+, B6.Cg-Gt(ROSA)26Sortm14(CAG-tdTomato)Hze/J | The Jackson Laboratory | Stock number 007914 RRID:IMSR_JAX:007914 | Male |
| Genetic reagent | AAV-Fos::CreERT2 | *Matos et al., 2019* | | Serotype 5 |
| Genetic reagent | AAV-hSyn::DIO-mCherry | *Matos et al., 2019* | | Serotype 5 |

*Continued on next page*

*Continued*

| Reagent type (species) or resource | Designation | Source or reference | Identifiers | Additional information |
|---|---|---|---|---|
| Antibody | Anti-Fos (rabbit) | Synaptic Systems | #226008 | 1:1000 |
| Antibody | Anti-Parvalbumin (mouse) | Chemicon/ Millipore | MAB1572, RRID:AB_2174013 | 1:2000 |
| Antibody | Anti-6E10 (mouse) | BioLegend | SIG-39320, RRID:AB_662798 | 1:1000 |
| Antibody | Anti-rabbit Alexa Fluor 488 (goat) | ThermoFisher Scientific | A-11008, RRID:AB_143165 | 1:400 |
| Antibody | Anti-mouse Alexa Fluor 405 (goat) | Invitrogen | A-31553, RRID:AB_221604 | 1:400 |
| Antibody | NeuroTrace 530/6115 Red fluorescent Nissl | ThermoFisher Scientific | N21483, RRID:AB_2572212 | 1:200 |
| Antibody | Anti-mouse 647 (goat) | Invitrogen | A-21245, RRID:AB_2535813 | 1:400 |
| Chemical compound, drug | Temgesic | RB Pharmaceuticals | | 0.1 mg/kg |
| Chemical compound, drug | Lidocaine | Sigma-Aldrich Chemie N.V. | | 2% |
| Chemical compound, drug | 4 -hydroxytamoxifen | HelloBio | HB6040 | 25 mg/kg, i.p. |
| Chemical compound, drug | DMSO | Sigma Aldrich Chemie N.V. | D8418 | |
| Chemical compound, drug | 2%Tween80 | Sigma Aldrich Chemie N.V. | P1754 | |
| Software, Algorithm | Ethovision XT | Noldus | | Video analysis |
| Software, algorithm | Fiji | ImageJ | Version 2.14.0/1.54 p | Image analysis |
| Software, algorithm | Graphpad Prism | GraphPad Software Inc | Version 10.4.1 | Analysis and data visualization |
| Software, algorithm | MATLAB | Mathworks | | Electrophysiology and image analysis |
| Software, algorithm | Python | Python Software Foundation | | Image analysis |
| Software, algorithm | IGOR Pro | WaveMetrics | Version 8.0 | Electrophysiology analysis |

## Animals

APP/PS1 mice (The Jackson Laboratory, APPswe, PSEN1de9, B6C3-Tg, stock number 004462) express a chimeric mouse/human APP gene harboring the Swedish double mutation K59N/M596L (APPswe) and a human PS1 gene harboring the exon 9 deletion (PS1dE9). Both are under the control of the mouse prion promoter (MoPrP.Xho) (*Jankowsky et al., 2004*; *Jankowsky et al., 2001*; *Jankowsky et al., 2003*). PV-Cre mice (The Jackson Laboratory, B6.129P2-Pvalbtm1(cre)Arbr/J, stock number 017320) express Cre recombinase under the control of the endogenous *Parvalbumin* promoter, directing Cre recombinase expression to mouse PV-expressing cells. SST-Cre mice (The Jackson Laboratory, Ssttm2.1(cre)Zjh/J, stock number 013044) express Cre recombinase under the control of the endogenous *Somatostatin* promoter, directing Cre recombinase expression to mouse SST-expressing cells. All mouse lines were maintained on a C57BL/6J background. Hemizygous APP/PS1 mice were crossed with hemizygous PV-Cre or SST-Cre mice to produce double-transgenic mice and single-transgenic controls. APP/PS1 PV-Cre-tdTomato mice were obtained by crossing APP/PS1 PV-Cre and wild type PV-Cre with R26AI14 mice (The Jackson Laboratory, B6.Cg-Gt(ROSA)26Sortm14(CAG-tdTomato)Hze/J, stock number 007914). Male mice were used in all experiments and were individually

housed on a 12 hr light/dark cycle with ad libitum access to water and food. All experiments were approved by the Netherlands Central Committee for Animal Experiments (CCD) and the Animal Ethical Care Committee (IVD) of the Vrije Universiteit Amsterdam.

## AAV vectors and stereotactic microinjections

AAV-Fos-CreER$^{T2}$ (titer: $1.2 \times 10^{13}$) and Cre-dependent AAV-hSyn-DIO-mCherry (titer: $1.68 \times 10^{11}$) were packaged as serotype 5 virus. Prior to stereotactic microinjections (*Matos et al., 2019*; *Van den Oever et al., 2013*), mice received Temgesic (0.1 mg/kg, RB Pharmaceuticals, UK) and were anesthetized using isoflurane. When mounted in a stereotactic frame, lidocaine (2%, Sigma-Aldrich Chemie N.V., The Netherlands) was applied topically to the skull for local analgesia. AAV mixtures of AAV-Fos-CreER$^{T2}$ and Cre-dependent AAV-hSyn-DIO-mCherry (ratio 1:500) were injected bilaterally in the mPFC (+1.8 mm AP; ±0.45 mm ML; −2.1 mm DV; relative to Bregma). For SST-Cre mice, the Cre-dependent AAV-hSyn-DIO-mCherry was injected. Each hemisphere was infused with 0.5 µl virus using microinjection glass needles, connected to a 10 µl Hamilton syringe by pressure ejection with a rate of 0.1 µl/min. This was followed by an additional 5 min to allow for diffusion of the viral mixture. Mice remained single-housed in their home-cage for 2 weeks until the start of behavioral experiments.

## Contextual fear memory

Four days prior to undergoing CFC, mice were handled for 2 consecutive days and then left undisturbed in their home. CFC training was performed in a soundproof cabinet with continuous white noise (68 dB) in a Plexiglas chamber with a stainless-steel grid floor (Ugo Basil, Italy). Between each trial, the CFC chamber was cleaned with 70% ethanol. At the beginning of each trial, the mice were allowed to explore the chamber for 120 s, followed by a foot shock (0.7 mA, 2 s). After 30 s, mice were returned to their home-cage. During memory retrieval tests, mice were re-exposed to the context and allowed to explore for 2 min. Freezing behavior was analyzed by video tracking using Ethovision XT (Noldus, the Netherlands) and was defined as a lack of movement for at least 1.5 s.

## 4TM treatment

Mice received 4TM (HB6040, HelloBio, 25 mg/kg, i.p.) 2 hr after CFC training. To make the 4TM aqueous solution, 15 mg 4TM was dissolved in 300 µl DMSO (D8418, Sigma-Aldrich Chemie N.V, The Netherlands). This solution was then further diluted with 2850 µl saline with 2% Tween 80 (P1754, Sigma-Aldrich Chemie N.V., The Netherlands) saline and then with 2850 µl saline, making a solution of 2.5 mg 4TM per ml saline, 5% DMSO, and 1% Tween 80 (*Matos et al., 2019*; *Ye et al., 2016*).

## Electrophysiological recordings

Mice were transcardially perfused with ice-cold partial sucrose solution (70 mM NaCl, 2.5 mM KCl, 1.25 NaH$_2$PO$_4$*H$_2$O, 5 mM MgSO$_4$*7H$_2$O, 1 mM CaCl$_2$*2H$_2$O, 70 mM sucrose, 25 mM D-glucose, 25 mM NaHCO$_3$, 1 mM sodium ascorbate, and 3 mM sodium pyruvate, carboxygenated with 5% CO$_2$/95% O$_2$, pH 7.4, 310 mOsm), followed by decapitation and rapid extraction of the brain. 300 µm coronal slices were made from the mPFC in ice-cold carboxygenated partial sucrose solution using a vibratome (Leica). After 15 min of incubation at 37°C in partial sucrose solution, slices were transferred to holding aCSF (125 mM NaCl, 3 mM KCl, 1.25 NaH$_2$PO$_4$*H$_2$O, 2 mM MgCl$_2$*6H$_2$O, 1.3 mM CaCl$_2$*2H$_2$O, 25 mM D-glucose, 25 mM NaHCO$_3$, 1 mM sodium ascorbate, and 3 mM sodium pyruvate, carboxygenated with 5% CO$_2$/95% O$_2$, pH 7.4, 310 mOsm). After a recovery step of 45 min at room temperature, slices were transferred to a recording chamber continuously perfused with carboxygenated running aCSF (holding aCSF without sodium ascorbate, sodium pyruvate, and only 1 mM MgCl$_2$*6H$_2$O, 34°C). The mPFC was identified using differential interference contrast microscopy, and PV cells expressing tdTomato or SST cells expressing mCherry were found using fluorescence. Pyramidal, PV, and SST cells were recorded in the mPFC using a Multiclamp 700B amplifier (Molecular Devices, Sunnyvale, CA, USA) and sampled at 10 kHz low-pass filter at 4 kHz and digitized with Axon Digidata 1440 A (Molecular Devices). Whole-cell recordings were performed using borosilicate glass electrodes (Science Products, Hofheim, Germany) with a tip resistance of 2–6 MOhm, filled with potassium-gluconate-based intracellular solution (148 mM K-gluconate, 1 mM KCl, 10 mM HEPES, 0.3 mM EGTA, 4 mM K$_2$-phosphocreatinine, 4 mM Mg-ATP, 0.4 mM GTP, adjusted to pH 7.3–7.4 with KOH, 300 mOsm). Once a gigaohm seal was acquired, a whole-cell configuration was obtained, and

the resting membrane potential was directly recorded. Passive and active membrane properties of PV, SST, and pyramidal neurons were measured in current clamp mode while being kept at –70 mV. Rheobase was analyzed using an incremental ramp-like injection of current from 0 pA to ±400 pA for 1200 ms. An input-output profile was generated by injecting incrementally increasing currents, starting at –100 pA to 250 pA in steps of 25 pA for pyramidal and SST cells, and up to 425 pA for PV cells. Cells with an access resistance of above 25 mOhm, as well as cells that showed unstable resting membrane potential or aberrant spiking patterns, were excluded from analysis. Data was analyzed using a custom-made script in MATLAB (Mathworks). PYR cell recordings at 20 weeks were acquired from APP PV Cre tdTomato and APP SST-Cre mice and control PV Cre tdTomato and SST-Cre, respectively.

Animals used for sEPSC and sIPSC recordings were immediately sacrificed after remote memory retrieval, similarly as described above. sEPSC and sIPSC recordings were recorded using a cesium gluconate-based intracellular (120 mM, 10 mM CsCl, 10 mM HEPES, 10 mM K-phosphocreatine, 2 ATP-Mg, 0.3 mM GTP, 0.2 mM EGTA, and 1 mM QX314, adjusted to pH 7.3–7.4 with CsOH). sEPSCs were recorded at a holding potential of –70 mV and sIPSCs were recorded at 0 mV. Pyramidal cells expressing mCherry were identified using fluorescence, and pyramidal cells not expressing mCherry were recorded in the same field of view and identified based on morphology. Access resistance was monitored during recording, and cells with an access resistance of above 20 mOhm were excluded. Cells with an unstable signal were excluded from analysis. Events were analyzed using TaroTools add-on in IgorPro 8.0 (WaveMetrics), where analysis was performed on 2 min of sEPSC and 1 min of sIPSC recordings.

## Immunohistochemistry

Mice were transcardially perfused with ice-cold PBS (pH 7.4), followed by perfusion with ice-cold 4% paraformaldehyde (PFA) in PBS (pH 7.4). After removing the brains and overnight post-fixation in 4% PFA solution, the brains were immersed in 30% sucrose PBS solution with 0.02% $NaN_3$. Coronal sections of 35 µm were made using a cryostat and were stored at 4°C in PBS with 0.02 $NaN_3$ until further use. Immunohistochemical staining was performed on free-floating brain sections as previously described (*Van den Oever et al., 2013*). The following antibodies were used: rabbit anti-Fos (1:1000, 226008, SySy), mouse anti-PV (1:2000, MAB1572, Chemicon/Millipore), goat anti-rabbit Alexa Fluor 488 (1:400, A11008, Thermo Fisher Scientific, USA), goat anti-mouse Alexa Fluor 405 (1:400, A31553, Invitrogen), and NeuroTrace 530/615 Red fluorescent Nissl (1:200, N21483, Thermo Fisher Scientific, USA). For the amyloid beta plaque quantification, mouse anti-6E10 (1:1000, SIG-39320, BioLegend) and goat anti-mouse 647 (1:400, A-21245, Invitrogen) were used. First, sections were washed with PBS and blocked for 1 hr at room temperature with 0.2% Triton X-100 and 5% fetal bovine serum in PBS. Primary antibodies were diluted in blocking solution, which was used to incubate sections overnight at 4°C. After primary antibody incubation, the sections were washed with PBS, followed by a 2 hr incubation step with the secondary antibody solution containing NeuroTrace Nissl at room temperature. Finally, sections were rinsed with PBS, mounted and coverslipped. To quantify the viral-TRAP immunostainings, images of the mPFC were acquired using a confocal microscope (Nikon, Eclipse Ti2), where 6–8 z-stacks were made per animal. The experimenter was blind to the genotype. Similar exposure time, gain settings, and camera settings were used in each set of experiments. Quantification of Nissl cells was done using ImageJ software, where regions of interest (ROIs) of neuronal cells were extracted. To exclude glial cells and nonspecific staining, we used a size range (80–2000 square units) and a circularity (0.5–1.0). MATLAB (Mathworks) was used to correct for cells included in multiple ROIs in sequential z-stacks. Cells positive for mCherry, Fos, or PV, and overlap thereof were counted manually using the ImageJ cell counter. To quantify PV labeling around mCherry$^+$ and mCherry$^-$ cells, we used a custom-made script in ImageJ to extract cell ROIs. The ROI was enlarged and used to measure the total amount of binarized PV labeling within the ring surrounding the cell. The total amount was normalized to the surface area of the ring. Using a custom-made script in MATLAB, data was categorized into mCherry$^+$ and neighboring mCherry$^-$-expressing neurons. For the beta-amyloid plaque dataset, images were acquired using a Carl Zeiss LSM 510 Meta confocal microscope. Per animal, eight z-stack images were made. Images were analyzed using an in-house built Python script to detect, count, and measure Aβ-plaques. Particles were included based on a circularity and size inclusion criterion. False positives were manually deleted from the dataset.

## Statistical analyses

Statistical details are presented in the figure legends, where the number of animals and the number of cells is shown. Statistical testing was done using GraphPad Prism (version 10.4.1, San Diego, CA, USA). D'Agostino and Pearson test was used to test for normality. When comparing two groups, a paired Student's $t$-test was used for groups with normally distributed data and a Mann-Whitney U test was used for groups with non-parametric data. For comparing observed (% of chance) levels with chance, a one-sample $t$-test was used. For the comparisons of multiple groups, a repeated-measures two-way ANOVA was used with a Bonferroni test for post hoc analysis. For the comparison of perisomatic PV labeling on mCherry$^+$ and neighboring mCherry$^-$ cells, we similarly used a repeated-measures two-way ANOVA to account for local variability in staining intensity. p-Values<0.05 were considered significant. Statistical outliers were identified using Grubbs' test. Data is shown as mean ± standard error of the mean (SEM). Experimenters were blind to the genotype of the mice to reduce bias when analyzing data.

## Acknowledgements

We thank Yvonne Gouwenberg for AAV packaging, Robbert Zalm for generating AAV constructs, and Tim Heijstek for technical assistance in electrophysiology. We also thank Nine Kok and Romina Ambrosini for assistance with immunohistochemical stainings. Ronald van Kesteren and Michel van den Oever received funding from the Dutch Alzheimer Association (Alzheimer Nederland; grant: WE.03-2020-05).

## Additional information

### Funding

| Funder | Grant reference number | Author |
|---|---|---|
| Alzheimer Nederland | WE.03-2020-05 | Ronald E van Kesteren |

The funders had no role in study design, data collection and interpretation, or the decision to submit the work for publication.

### Author contributions

Julia J van Adrichem, Conceptualization, Data curation, Formal analysis, Validation, Investigation, Visualization, Methodology, Writing – original draft, Writing – review and editing; Rolinka J van der Loo, Romina Ambrosini Defendi, Investigation, Methodology; August B Smit, Supervision, Writing – review and editing; Michel C van den Oever, Ronald E van Kesteren, Conceptualization, Formal analysis, Supervision, Funding acquisition, Project administration, Writing – review and editing

### Author ORCIDs

Julia J van Adrichem (ID) https://orcid.org/0000-0002-9771-5197
Michel C van den Oever (ID) https://orcid.org/0000-0001-5523-8612
Ronald E van Kesteren (ID) https://orcid.org/0000-0002-9592-3777

### Ethics

All experiments were approved by the Netherlands Central Committee for Animal Experiments (CCD) and the animal ethical care committee (IVD) of the Vrije Universiteit Amsterdam.

Reviewer #2 (Public review): https://doi.org/10.7554/eLife.106866.3.sa1
Author response https://doi.org/10.7554/eLife.106866.3.sa2

## Additional files

### Supplementary files

Supplementary file 1. Supplementary tables describing parvalbumin (PV) and pyramidal (PYR) cell

properties at 16 weeks and 20 weeks, as well as somatostatin (SST) cell properties at 20 weeks.

MDAR checklist

Source data 1. Individual datapoints of parvalbumin (PV) cell properties at 16 weeks in *Supplementary file 1, table 1A*.

Source data 2. Individual datapoints of pyramidal (PYR) cell properties at 16 weeks in *Supplementary file 1, table 1B*.

Source data 3. Individual datapoints of parvalbumin (PV) cell properties at 20 weeks in *Supplementary file 1, table 1C*.

Source data 4. Individual datapoints of pyramidal (PYR) cell properties at 20 weeks in *Supplementary file 1, table 1D*.

Source data 5. Individual datapoints of somatostatin (SST) cell properties at 20 weeks in *Supplementary file 1, table 1E*.

## Data availability

All data generated or analysed in this study are included in this article and in the source data file. Further information can be requested from Ronald E. van Kesteren (ronald.van. kesteren@vu.nl) or Michel C. van den Oever (michel.vanden.oever@vu.nl). The custom analysis scripts developed for this study were deposited on GitHub. Plaque detection: https://github.com/revankes/plaque-detection (commit 7a7d5f8; copy archived at *Defendi, 2025*); electrophysiology analysis: https://github.com/revankes/Electrophysiology-analysis (commit 51d59f6; copy archived at *Adrichem, 2025*); cell detection: https://github.com/revankes/Cell-detection (commit c55d116; copy archived at *Gebuis, 2025*); perisomatic labelling: https://github.com/revankes/Perisomatic-labeling-detection (commit8df42ff; copy archived at *Adrichem and Gebuis, 2025*). All codes were released under the MIT license.

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
