## [Editor Report · eLife Assessment]

This study provides **valuable** insights into the mechanisms of remote memory impairment in an Alzheimer's disease mouse model. The evidence is **compelling**, with careful use of viral-TRAP labeling and patch-clamp electrophysiology to demonstrate altered inhibitory microcircuit function, though the mechanistic link to memory deficits remains correlative. Overall, the work advances understanding of early circuit-level changes in AD, while highlighting open questions regarding causality and broader network contributions.

---

## [Referee Report · Reviewer #2 (Public review)]

This study presents a thorough investigation of remote memory deficits in the APP/PS1 mouse model of Alzheimer's disease, highlighting the progressive emergence of these deficits alongside selective hyperexcitability of PV interneurons in the mPFC. By combining viral-TRAP labeling and patch-clamp electrophysiology, the authors demonstrate increased inhibitory input onto engram cells in APP/PS1 mice, despite preserved engram size and reactivation. The revised manuscript successfully addresses earlier concerns by clarifying the relationship between amyloid pathology and circuit dysfunction, acknowledging the correlative nature of the findings, and integrating possible contributions of excitatory remodeling and broader network changes, including oscillatory disruptions. Although the precise mechanistic link between PV hyperexcitability, increased inhibition, and impaired remote memory remains to be empirically established, the study convincingly argues for inhibitory microcircuit alterations as an early contributor to cognitive decline in AD.

---

## [Author Response]

The following is the authors’ response to the original reviews

**Public Reviews:**

**Reviewer #1 (Public review):**
This study presents evidence that remote memory in the APP/PS1 mouse model of Alzheimer's disease (AD) is associated with PV interneuron hyperexcitability and increased inhibition of cortical engram cells. Its strength lies in the fact that it explores a neglected aspect of memory research - remote memory impairments related to AD (for which the primary research focus is usually on recent memory impairments) -which has received minimal attention to date. While the findings are intriguing, the weakness of the paper hovers around purely correlational types of evidence and superficial data analyses, which require substantial revisions as outlined below.

We thank the reviewer for their feedback, and we appreciate the recognition of the study’s novelty in addressing remote memory impairments in AD. We acknowledge the reviewer’s concerns and have implemented revisions to strengthen the manuscript.

Major concerns:(1) In light of previous work, including that by the authors themselves, the data in Figure 1 should be implemented by measurements of recent memory recall in order to assess whether remote memories are exclusively impaired or whether remote memory recall merely represents a continuation of recent memory impairments.

We agree with the reviewer that is an important point. In line with their suggestion in minor comment 1, we now omitted the statement on recent memory in the results (previously on lines 109-111 and 117). Nonetheless, previous independent experiments from our group have repeatedly shown recent memory deficits in APP/PS1 mice at 12 weeks of age, including a recent article published in 2023. We refer the reviewer to figure 2c in Végh et al. (2014) and figure 2i in Kater et al. (2023). We have added a reference of the latter paper to our discussion section (line 458-459). Therefore, we are confident that the recent memory deficit at 12 weeks of age is a stable phenotype in our APP/PS1 mice.

With these data in mind, we argue that the remote memory recall impairment is not a continuation of recent memory impairments. Recent memory deficits emerge already at 12 weeks of age, and when remote memory is assessed at 16 weeks (4 weeks after training at 12 weeks of age), APP/PS1 mice are still capable of forming and retrieving a remote memory. This suggests that remote memory retrieval can occur even when recent memory is compromised, arguing against the idea that the remote memory deficit observed at 20 weeks is a continuation of earlier recent memory impairments. We have clarified this point in the revised manuscript by adding the following sentence to the discussion section (line 462-465):

‘This suggests that a remote memory can be formed even when recent memory expression is already compromised, indicating that the remote memory deficit in 20-week-old APP/PS1 mice is not a continuation of earlier recent memory impairments.’

(2) Figure 2 shows electrophysiological properties of PV cells in the mPFC that correlate with the behavior shown in Figure 1. However, the mice used in Figure 2 are different than the mice used in Figure 1. Thus, the data are correlative at best, and the authors need to confirm that behavioral impairments in the APP/PS1 mice crossed to PV-Cre (and SST-Cre mice) used in Figure 2 are similar to those of the APP/PS1 mice used in Figure 1. Without that, no conclusions between behavioral impairments and electrophysiological as well as engram reactivation properties can be made, and the central claims of the paper cannot be upheld.

We thank the reviewer for raising this concern. Indeed, the remote memory impairment and PV hyperexcitability are correlative data, and therefore we do not make causal claims based on these data. However, please note that most of our key findings, including behavioural impairments, characterization of the engram ensemble and reactivation thereof, as well as inhibitory input measurements, were acquired using the same mouse line (APP/PS1), strengthening the coherence of our conclusions. Also, our electrophysiological findings in APP/PS1 (enhanced sIPSC frequency) and APP/PS1-PV-Cre-tdTomato (enhanced PV cell excitability) mice align well. Direct comparisons between the transgenic mouse lines APP/PS1 and APP/PS1 Parv-Cre were performed in our previous studies, confirming that these lines are similar in terms of behaviour and pathology. Specifically, we demonstrated that APP/PS1 mice display spatial memory impairments at 16 weeks of age, Fig 4a-d, consistent with the deficits observed in APP/PS1 Parv-Cre mice at 16 weeks of age, Fig 5a-c (Hijazi et al., 2020a). Additionally, Hijazi et al. (2020a) showed that soluble and insoluble Aβ levels do not differ between APP/PS1 Parv-Cre and APP/PS1 mice (sFig. 1), indicating comparable levels of pathology between these lines. While we do not have a similar characterization of the APP/PS1 SST-Cre line, we should mention that we also did not observe excitability differences in SST cells. We now acknowledge the limitation in the revised discussion section (line 480-487), and stress that our electrophysiology and behavioural findings are correlative in nature:

‘Although the excitability measurements were performed in APP/PS1-PV-Cre-tdTomato mice, and not in the APP/PS1 parental line, we previously found that these transgenic mouse lines exhibit comparable amyloid pathology (both soluble and insoluble amyloid beta levels) as well as similar spatial memory deficits (Hijazi et al., 2020a; Kater et al., 2023). Thus, our observations indicate that the APP/PS1 PV-Cre-tdTomato and APP/PS1 lines are similar in terms of pathology and behaviour. Nonetheless, further work is needed to identify a causal link between PV cell hyperexcitability and remote memory impairment.’

(3) The reactivation data starting in Figure 3 should be analysed in much more depth:a) The authors restrict their analysis to intra-animal comparisons, but additional ones should be performed, such as inter-animal (WT vs APP/PS1) as well as inter-age (12-16w vs 16-20w). In doing so, reactivation data should be normalized to chance levels per animal, to account for differences in labelling efficiency - this is standard in the field (see original Tonegawa papers and for a reference). This could highlight differences in total reactivation that are already apparent, such as for instance in WT vs APP/PS1 at 20w (Figure 3o) and highlight a decrease in reactivation in AD mice at this age, contrary to what is stated in lines 213-214.

We would like to thank the reviewer for the valuable input on the reactivation data in Figure 3.

We agree with the reviewer and now depict the data as normalized to chance levels (Figure 3). The original figures are now supplemental (sFig. 5). The reactivation data normalized to chance are similar to the original results, i.e. no difference was observed in the reactivation of the mPFC engram ensemble between genotypes. The reviewer may have overlooked that we did perform inter-animal (WT vs. APP/PS1) comparisons, however these were not significantly different. We have made this clearer in the main text, lines 277, 288-289, 294-295 and 303-304. Moreover, the reviewer recommended including inter-age group comparisons, which have now been added to the supplemental figures (sFig. 6). No genotype-dependent differences were observed. While a main effect of age group did emerge, indicating that there is a potential increased overlap between Fos+ and mCherry+ in animals aged 16-20 weeks, we caution against overinterpreting this finding. These experimental groups were processed in separate cohorts, with viral injection and 4TM-induced tagging performed at different moments in time, which may have contributed to the observed differences in overlap. We have addressed this point in the revised discussion (line 612-617):

‘Furthermore, we also observed an increase in the amount overlap between Fos+ and mCherry+ engram cells when comparing the 12-16w and 16-20w age groups. This finding should be interpreted with caution, as the experimental groups were processed in separate cohorts, with viral injections and 4TM-induced tagging performed at different moments in time. This may have contributed to the observed differences between ages.’

b) Comparing the proportion of mcherry+ cells in PV- and PV+ is problematic, considering that the PV- population is not "pure" like the PV+, but rather likely to represent a mix of different pyramidal neurons (probably from several layers), other inhibitory neurons like SST and maybe even glial cells. Considering this, the statement on line 218 is misleading in saying that PVs are overrepresented. If anything, the same populations should be compared across ages or groups.

We thank the reviewer for their insightful comment and agree that the PV- population of cells is likely more heterogenous than the PV+ population. However, we would like to clarify that all quantified cells were selected based on Nissl immunoreactivity, and to exclude non-neuronal cells, stringent thresholding was applied in the script that was used to identify Nissl+ cells. The threshold information has now been added to the methods section (line 758-760). Thus, although heterogenous, the analysed PV- population reflects a neuronal subset. In response to the reviewer’s suggestion, we have now included overlap measurements relative to chance levels (Figure 3). These analyses did not reveal differences with the original analyses, i.e., there are no genotype specific differences. We have also incorporated the suggested inter-age group comparisons (sFig. 6) and found no differences between age groups. In light of the raised concerns, we have removed the statement that PV cells were overrepresented in the engram ensemble.

c) A similar concern applies to the mcherry- population in Figure 4, which could represent different types of neurons that were never active, compared to the relatively homogeneous engram mcherry+ population. This could be elegantly fixed by restricting the comparison to mCherry+Fos+ vs mCherry+Fos- ensembles and could indicate engram reactivation-specific differences in perisomatic inhibition by PV cells.

The comparison the reviewer suggests, comparing mCherry+Fos+ to mCherry+Fos- is indeed conceptually interesting and could provide more insight into engram reactivation and PV input. However, there are practical limitations to performing this analysis, as neurons in close proximity need to be compared in a pairwise manner to account for local variability in staining intensity. As shown in Figure 3c+k and Figure 4a+b, d+e, PV immunostaining intensity varies to a certain extend within a given image. While pairwise comparisons of neighbouring neurons were feasible when analysing mCherry+ and mCherry- cells, they are unfortunately not feasible for the mCherry+Fos+ vs. mCherry+Fos- comparison. The occurrence of spatially adjacent mCherry+Fos+ and mCherry+Fos- neurons is too sparse for a pairwise comparison. This analysis would therefore result in substantial under-sampling and limit the reliability of the analysis. Nonetheless, we agree with the reviewer that the mCherry- population may be more heterogenous than the mCherry+ population, despite the fact that PV+ neurons and that non-neuronal cells were excluded from both populations in the analyses. We therefore added a statement to the discussion to acknowledge this limitation (line 536-539):

‘Although PV+ cells were not included in this analysis and we excluded non-neuronal cells based on the area of the Nissl stain, the mCherry- population was potentially more heterogenous than the mCherry+ population, which may have contributed to the differences we observed.’

(4) At several instances, there are some doubts about the statistical measures having been employed:a) In Figure 4f, it is unclear why a repeated measurement ANOVA was used as opposed to a regular ANOVA.b) In Supplementary Figure 2b, a Mann-Whitney test was used, supposedly because the data were not normally distributed. However, when looking at the individual data points, the data does seem to be normally distributed. Thus, the authors need to provide the test details as to how they measured the normalcy of distribution.

a) Based on the pairwise comparison of neighbouring neurons within animals, the data in Figure 4f was analysed with a repeated measure ANOVA.

b) We thank the author for their comment on Supplementary Figure 2b. The data is indeed normally distributed, and we have analysed it using a D’Agostino & Pearson test. We have corrected this in the supplemental figure.

Minor concerns:(1) Line 117: The authors cite a recent memory impairment here, as shown by another paper. However, given the notorious difficulty in replicating behavioral findings, in particular in APP/PS1 mice (number of backcrossings, housing conditions, etc., might differ between laboratories), such a statement cannot be made. The authors should either show in their own hands that recent memory is indeed affected at 12 weeks of age, or they should omit this statement.

We thank the reviewer for this thoughtful comment. As noted in our response to major concern (1), we have addressed this concern by providing additional information and clarification in the discussion (line 462-465) regarding the possibility that remote memory impairments are a continuation of recent memory impairments. As mentioned in our response, we have added a reference to a more recent study from our lab (Kater et al. 2023). These findings are consistent with the earlier report from our lab (Végh et al. 2014), underscoring the reproducibility of this phenotype across independent cohorts and time. Notably, the experiments in the 2023 and present study were performed using the same housing and experimental conditions. Nevertheless, in light of the reviewer’s suggestion, and to avoid overstatement or speculation, we have now omitted the sentence referring to recent memory impairments at 12 weeks of age from the results section.

(2) Pertaining to Figure 3, low-resolution images of the mPFC should be provided to assess the spread of injection and the overall degree of double-positive cells.

We agree with the reviewer and have added images of the mPFC as a supplemental figure (sFig. 3) that show the spread of the injection. Unfortunately, it is not possible to visualize the overall degree of double-positive cells at a lower magnification (or low-resolution). Representative examples of colocalization are presented in Figure 3.

**Reviewer #2 (Public review):**
This study presents a comprehensive investigation of remote memory deficits in the APP/PS1 mouse model of Alzheimer's disease. The authors convincingly show that these deficits emerge progressively and are paralleled by selective hyperexcitability of PV interneurons in the mPFC. Using viral-TRAP labeling and patch-clamp electrophysiology, they demonstrate that inhibitory input onto labeled engram cells is selectively increased in APP/PS1 mice, despite unaltered engram size or reactivation. These findings support the idea that alterations in inhibitory microcircuits may contribute to cognitive decline in AD.However, several aspects of the study merit further clarification. Most critically, the central paradox, i.e., increased inhibitory input without an apparent change in engram reactivation, remains unresolved. The authors propose possible mechanisms involving altered synchrony or impaired output of engram cells, but these hypotheses require further empirical support. Additionally, the study employs multiple crossed transgenic lines without reporting the progression of amyloid pathology in the mPFC, which is important for interpreting the relationship between circuit dysfunction and disease stage. Finally, the potential contribution of broader network dysfunction, such as spontaneous epileptiform activity reported in APP/PS1 mice, is also not addressed.

We thank the reviewer for their evaluation and appreciate the positive assessment of our study’s contributing to understanding remote memory deficits and the dysfunction of inhibitory microcircuits in AD. We also acknowledge the relevant points raised and have revised the manuscript to clarify our interpretations.

**Recommendations for the authors:**

**Reviewer #1 (Recommendations for the authors):**
(1) Line 68: What are "APP23xPS45" mice? This is most likely a typo.

This line is a previously reported double transgenic amyloid beta mouse model that was obtained by crossing APP23 (overexpressing human amyloid precursor protein with the Swedish double mutation at position 670/671) with PS45 (carrying a transgene for mutant Presenilin 1, G384A mutation) (Busche et al., 2008; Grienberger et al., 2012).

(2) Line 148: The authors should also briefly describe in the main text that APP/PS1 x SST-Cre mice were generated and used here.

We thank the reviewer for their comment and have added their suggestion to the main text (line 166-168):

‘To do this, APP/PS1 mice were crossed with SST-Cre mice to generate APP/PS1 SST-Cre mice. Following microinjection of AAV-hSyn::DIO-mCherry into the mPFC, recordings were obtained from SST neurons.’

(3) The discussion should be condensed because of redundancies on several occasions. For example, memory allocation is discussed starting on line 371, then again on line 392. This should be combined. Likewise, how the correlative nature of the findings about PV interneurons could be further functionally addressed is discussed on lines 413 and 454, and should be condensed into one paragraph.

We thank the reviewer for this suggestion and have revised the discussion to remove the redundancies as proposed.

**Reviewer #2 (Recommendations for the authors):**
To strengthen the manuscript, the following points should be addressed:(1) Quantify amyloid pathology: It is essential to assess amyloid-β levels (soluble and insoluble) in the mPFC of APP/PS1-PV-Cre-tdTomato mice at the studied ages. This would help determine whether the observed circuitlevel changes track with disease progression as seen in canonical APP/PS1 models.

We thank the reviewer for this valuable suggestion and agree that assessing Aβ levels in the mPFC is important to determine whether the observed circuit level alterations in APP/PS1 mice coincide with the progression of amyloid pathology. Therefore, we assessed the amyloid plaque load in the mPFC of APP/PS1 mice at 16 and 20 weeks of age (new supplemental figure sFig. 1) and observed no difference in plaque load between these two time points. This suggests that the increased excitability in the mPFC cannot be attributed to differences in plaque load (insoluble amyloid beta).

In line with this, we previously studied both soluble and insoluble Aβ levels in the CA1 and reported that there are no differences between 12 and 16 weeks of age (Kater et al., 2023), while PV cell hyperexcitability is present at 16 weeks of age (Hijazi et al., 2020a). From 24 weeks onwards, the level of amyloid beta increases. Similarly, Végh et al. (2014) showed using immunoblotting that monomeric and low molecular weight oligomeric forms of soluble Aβ are already present as early as 6 weeks of age and become more prominent at 24 weeks of age. Although the soluble Aβ measurements were performed in the hippocampus, we think these findings can be extrapolated to cortical regions, as the APP and PS1 mutations in APP/PS1 mice are driven by a prion promotor, which should induce consistent expression across brain regions. Data from other research groups support this hypothesis (Kim et al., 2015; Zhang et al., 2011). Thus, large regional differences in soluble Aβ are not expected. The temporal progression suggests that increasing levels of soluble amyloid beta might contribute to the emergence of PV cell hyperexcitability. We have added this point to the manuscript (line 585-591):

‘Since amyloid beta plaque load in the mPFC remains comparable between 16- and 20-week-old APP/PS1 mice, the observed increased excitability is unlikely the result of changes in insoluble amyloid beta levels. Previous data from our lab show that soluble amyloid beta is already present as early as 6 weeks of age and becomes more prominent at 24 weeks of age (Kater et al., 2023; Végh et al., 2014). The progressive increase in soluble amyloid beta levels may contribute to the emergence of PV cell hyperexcitability.’

Finally, we previously compared soluble and insoluble amyloid beta levels in APP/PS1 and APP/PS1 Parv Cre mice and show that these are similar (Hijazi et al., 2020a). While our current study shows the progression of amyloid beta accumulation in APP/PS1 mice, these mice also exhibit altered microcircuitry (enhanced sIPSC frequency on engram cells) at 20 weeks of age, the same age at which we observed PV cell hyperexcitability in APP/PS1 Parv Cre tdTomato mice. This further supports the generalizability of our findings across genotypes, between APP/PS1 and APP/PS1 Parv Cre tdTomato mice.

(2) Examine later disease stages: Since the current effects are modest, assessing memory performance, PV cell excitability, and engram inhibition at more advanced stages could clarify whether these alterations become more pronounced with disease progression.

We thank the reviewer for this thoughtful suggestion. Investigating advanced disease stages could indeed provide valuable insights into whether the observed alterations in memory performance, PV cell hyperexcitability and engram inhibition become more pronounced over time. Our previous work has shown that changes in pyramidal cell excitability emerge at a later stage than in PV cells, supporting the idea of progressive circuit dysfunction (Hijazi et al., 2020a). However, at these more advanced stages, additional pathological processes, such as an increased gliosis (Janota, Brites, Lemere, & Brito, 2015; Kater et al., 2023) and synaptic loss (Alonso-Nanclares, MerinoSerrais, Gonzalez, & DeFelipe, 2013; Bittner et al., 2012), will likely contribute to both electrophysiological and behavioural measurements. Furthermore, we would like to point out that the current changes observed in memory performance, PV hyperexcitability and increased inhibitory input on engram cells at 16-20 weeks of age are not modest, but already quite substantial. Our focus on these early time points in APP/PS1 mice were intentional, as it helps us understand the initial changes in Alzheimer’s disease at a circuit level and to identify therapeutic targets early intervention. What happens at later stages is certainly of interest, but beyond the scope of this study and should therefore be addressed in future studies. We have incorporated a discussion related to this point into the revised manuscript (line 602-606):

‘Moreover, it is relevant to investigate whether changes in PV and PYR cell excitability, as well as input onto engram cells in the mPFC, become more pronounced at later disease stages. Nonetheless, by focussing on early disease timepoints in the present study, we aimed to understand the initial circuit-level changes in AD and identify targets for early therapeutic intervention.’

(3) Address network hyperexcitability: Spontaneous epileptiform activity has been reported in APP/PS1 mice from 4 months of age (Reyes-Marin & Nuñez, 2017). Including EEG data or discussing this point in relation to your findings would help contextualize the observed inhibitory remodeling within broader network dysfunction.

We thank the reviewer for this valuable input and for highlighting the study by Reyes-Marin and Nuñez (2017). In line with this, we recently reported longitudinal local field potential (LFP) recordings in freely behaving APP/PS1 Parv-Cre mice and wild type control animals between the ages of 3 to 12 months (van Heusden et al., 2023). Weekly recordings were performed in the home cage under awake mobile conditions. These data showed no indications of epileptiform activity during wakefulness, consistent with previous findings that epileptic discharges in APP/PS1 mice predominantly occur during sleep (Gureviciene et al., 2019). Recordings were obtained from the prefrontal cortex (PFC), parietal cortex and the hippocampus. In contrast, the study by Reyes-Marin and Nuñez (2017) recorded from the somatosensory cortex in anesthetized animals. Here, during spontaneous recordings, no differences were observed in delta, theta or alpha frequency bands between APP/PS1 and WT mice. Interestingly, we observed an early increase in absolute power, particularly in the hippocampus and parietal cortex from 12 to 24 weeks of age in APP/PS1 mice. In the PFC we found a shift in relative power from lower to higher frequencies and a reduction in theta power. Connectivity analyses revealed a progressive, age-dependent decline in theta/alpha coherence between the PFC and both the parietal cortex and hippocampus. Given the well-established role of PV interneurons network synchrony and coordinating theta and gamma oscillations critical for cognitive function (Sohal, Zhang, Yizhar, & Deisseroth, 2009; Xia et al., 2017), these findings support the idea of early circuit dysfunction in APP/PS1 mice. Our findings, i.e. hyperexcitability of PV cells, align with these LFP based networklevel observations. These data suggest an early shift in the E/I balance, contributing to altered oscillatory dynamics and impaired inter-regional connectivity, possibly leading to alterations in memory. However, whether the observed PV hyperexcitability in our study directly contributes to alterations in power and synchrony remains to be elucidated. Furthermore, it would be interesting to determine the individual contribution of PV cell hyperexcitability in the hippocampus versus the mPFC to network changes and concurrent memory deficits. We have added a statement on network hyperexcitability to the discussion (line 561-565).

‘Interestingly, we recently found a progressive disruption of oscillatory network synchrony between the mPFC and hippocampus in APP/PS1 Parv-Cre mice (van Heusden et al., 2023). However, whether the observed PV cell hyperexcitability directly contributes to changes in inter-regional synchrony, and whether this leads to alterations at a network level, i.e. increased inhibitory input on engram cells, and consequently to memory deficits, remains to be elucidated in future studies.’

(4) Mechanisms responsible for PV hyperexcitability: Related to the previous point, a discussion of the possible underlying mechanisms, e.g., direct effects of amyloid-β, inflammatory processes, or compensatory mechanisms, would strengthen the discussion.

We agree with the reviewer that this will strengthen the discussion. We have now added a comprehensive discussion in the revised manuscript to address potential mechanisms responsible for PV cell hyperexcitability (line 579-594).:

‘Prior studies have shown that neurons in the vicinity of amyloid beta plaques show increased excitability (Busche et al., 2008). We demonstrated that PV neurons in the CA1 are hyperexcitable and that treatment with a BACE1 inhibitors, i.e. reducing amyloid beta levels, rescues PV excitability (Hijazi et al., 2020a). In line with this, we also reported that addition of amyloid beta to hippocampal slices increases PV excitability, without altering pyramidal cell excitability (Hijazi et al., 2020a). Finally, applying amyloid beta to an induced mouse model of PV hyperexcitability further impairs PV function (Hijazi et al., 2020b). Since amyloid beta plaque load in the mPFC remains comparable between 16- and 20-week-old APP/PS1 mice, the observed increased excitability is unlikely the result of changes in insoluble amyloid beta levels. Previous data from our lab show that soluble amyloid beta is already present as early as 6 weeks of age and becomes more prominent at 24 weeks of age (Kater et al., 2023; Végh et al., 2014). The progressive increase in soluble amyloid beta levels may contribute to the emergence of PV cell hyperexcitability. We hypothesize that the hyperexcitability induced by amyloid beta may result from disrupted ion channel function, as PV neuron dysfunction can result from altered potassium (Olah et al., 2022) and sodium channel activity (Verret et al., 2012).’

(5) Excitatory-inhibitory balance: While the main focus is on increased inhibition onto engram cells, the reported increase in sEPSC frequency (Figure 5g) across genotypes suggests the presence of excitatory remodelling as well. A brief discussion of how this may interact with increased inhibition would be valuable.

We thank the reviewer for this comment regarding the interaction between excitatory and inhibitory remodelling. We have now incorporated this discussion point into the revised manuscript (line 528-534):

‘Interestingly, both WT and APP/PS1 mice showed an increase in sEPSC frequency onto engram cells, suggesting that increased excitatory input is a consequence of memory retrieval and not affected by genotype. However, only in APP/PS1 mice, the augmented excitatory input coincided with an elevation of inhibitory input onto engram cells. The resulting imbalance between excitation and inhibition could therefore potentially disrupt the precise control of engram reactivation and contribute to the observed remote memory impairment.’

References

Alonso-Nanclares, L., Merino-Serrais, P., Gonzalez, S., & DeFelipe, J. (2013). Synaptic changes in the dentate gyrus of APP/PS1 transgenic mice revealed by electron microscopy. J Neuropathol Exp Neurol, 72(5), 386-395. doi:10.1097/NEN.0b013e31828d41ec

Bittner, T., Burgold, S., Dorostkar, M. M., Fuhrmann, M., Wegenast-Braun, B. M., Schmidt, B., . . . Herms, J. (2012). Amyloid plaque formation precedes dendritic spine loss. Acta Neuropathologica, 124(6), 797807. doi:10.1007/s00401-012-1047-8

Busche, M. A., Eichhoff, G., Adelsberger, H., Abramowski, D., Wiederhold, K. H., Haass, C., . . . Garaschuk, O. (2008). Clusters of hyperactive neurons near amyloid plaques in a mouse model of Alzheimer's disease. Science, 321(5896), 1686-1689. doi:10.1126/science.1162844

Grienberger, C., Rochefort, N. L., Adelsberger, H., Henning, H. A., Hill, D. N., Reichwald, J., . . . Konnerth, A. (2012). Staged decline of neuronal function in vivo in an animal model of Alzheimer's disease. Nat Commun, 3, 774. doi:10.1038/ncomms1783

Gureviciene, I., Ishchenko, I., Ziyatdinova, S., Jin, N., Lipponen, A., Gurevicius, K., & Tanila, H. (2019). Characterization of Epileptic Spiking Associated With Brain Amyloidosis in APP/PS1 Mice. Front Neurol, 10, 1151. doi:10.3389/fneur.2019.01151

Hijazi, S., Heistek, T. S., Scheltens, P., Neumann, U., Shimshek, D. R., Mansvelder, H. D., . . . van Kesteren, R. E. (2020a). Early restoration of parvalbumin interneuron activity prevents memory loss and network hyperexcitability in a mouse model of Alzheimer's disease. Mol Psychiatry, 25(12), 3380-3398. doi:10.1038/s41380-019-0483-4

Hijazi, S., Heistek, T. S., van der Loo, R., Mansvelder, H. D., Smit, A. B., & van Kesteren, R. E. (2020b). Hyperexcitable Parvalbumin Interneurons Render Hippocampal Circuitry Vulnerable to Amyloid Beta. iScience, 23(7), 101271. doi:10.1016/j.isci.2020.101271

Janota, C. S., Brites, D., Lemere, C. A., & Brito, M. A. (2015). Glio-vascular changes during ageing in wild-type and Alzheimer's disease-like APP/PS1 mice. Brain Res, 1620, 153-168. doi:10.1016/j.brainres.2015.04.056

Kater, M. S. J., Huffels, C. F. M., Oshima, T., Renckens, N. S., Middeldorp, J., Boddeke, E., . . . Verheijen, M. H. G. (2023). Prevention of microgliosis halts early memory loss in a mouse model of Alzheimer's disease. Brain Behav Immun, 107, 225-241. doi:10.1016/j.bbi.2022.10.009

Kim, H. Y., Kim, H. V., Jo, S., Lee, C. J., Choi, S. Y., Kim, D. J., & Kim, Y. (2015). EPPS rescues hippocampus-dependent cognitive deficits in APP/PS1 mice by disaggregation of amyloid-β oligomers and plaques. ature Communications, 6(1), 8997. doi:10.1038/ncomms9997

Olah, V. J., Goettemoeller, A. M., Rayaprolu, S., Dammer, E. B., Seyfried, N. T., Rangaraju, S., . . . Rowan, M. J. M. (2022). Biophysical Kv3 channel alterations dampen excitability of cortical PV interneurons and contribute to network hyperexcitability in early Alzheimer’s. Elife, 11, e75316. doi:10.7554/eLife.75316

Reyes-Marin, K. E., & Nuñez, A. (2017). Seizure susceptibility in the APP/PS1 mouse model of Alzheimer's disease and relationship with amyloid β plaques. Brain Res, 1677, 93-100. doi:10.1016/j.brainres.2017.09.026

Sohal, V. S., Zhang, F., Yizhar, O., & Deisseroth, K. (2009). Parvalbumin neurons and gamma rhythms enhance cortical circuit performance. Nature, 459(7247), 698-702. doi:10.1038/nature07991

van Heusden, F. C., van Nifterick, A. M., Souza, B. C., França, A. S. C., Nauta, I. M., Stam, C. J., . . . van Kesteren, R. E. (2023). Neurophysiological alterations in mice and humans carrying mutations in APP and PSEN1 genes. Alzheimers Res Ther, 15(1), 142. doi:10.1186/s13195-023-01287-6

Végh, M. J., Heldring, C. M., Kamphuis, W., Hijazi, S., Timmerman, A. J., Li, K. W., . . . van Kesteren, R. E. (2014). Reducing hippocampal extracellular matrix reverses early memory deficits in a mouse model of Alzheimer's disease. Acta Neuropathol Commun, 2, 76. doi:10.1186/s40478-014-0076-z

Verret, L., Mann, E. O., Hang, G. B., Barth, A. M., Cobos, I., Ho, K., . . . Palop, J. J. (2012). Inhibitory interneuron deficit links altered network activity and cognitive dysfunction in Alzheimer model. Cell, 149(3), 708-721. doi:10.1016/j.cell.2012.02.046

Xia, F., Richards, B. A., Tran, M. M., Josselyn, S. A., Takehara-Nishiuchi, K., & Frankland, P. W. (2017). Parvalbumin-positive interneurons mediate neocortical-hippocampal interactions that are necessary for memory consolidation. Elife, 6. doi:10.7554/eLife.27868

Zhang, W., Hao, J., Liu, R., Zhang, Z., Lei, G., Su, C., . . . Li, Z. (2011). Soluble Aβ levels correlate with cognitive deficits in the 12-month-old APPswe/PS1dE9 mouse model of Alzheimer's disease. Behavioural Brain Research, 222(2), 342-350. doi:https://doi.org/10.1016/j.bbr.2011.03.072